# DRAX: SPEECH RECOGNITION WITH DISCRETE FLOW MATCHING

## ABSTRACT

Diffusion and flow-based non-autoregressive (NAR) models have shown strong promise in large language modeling, however, their potential for automatic speech recognition (ASR) remains largely unexplored. We propose Drax, a discrete flow matching framework for ASR that enables efficient parallel decoding. To better align training with inference, we construct an audio-conditioned probability path that guides the model through trajectories resembling likely intermediate inference errors, rather than direct random noise to target transitions. Our theoretical analysis links the generalization gap to divergences between training and inference occupancies, controlled by cumulative velocity errors, thereby motivating our design choice. Empirical evaluation demonstrates that our approach attains recognition accuracy on par with state-of-the-art speech models while offering improved accuracy-efficiency trade-offs, highlighting discrete flow matching as a promising direction for advancing NAR ASR.

## 1 INTRODUCTION

Automatic speech recognition (ASR) has become a core component of modern machine learning systems, enabling speech-based interfaces, multilingual communication, and accessibility applications. Recent progress has been driven by large-scale autoregressive (AR) encoder-decoder models such as Whisper (Radford et al., 2023) and Qwen2-Audio (Chu et al., 2024), which achieve remarkable accuracy across languages and domains. However, the sequential nature of AR decoding creates an inherent efficiency bottleneck: tokens must be generated one by one, resulting in inference latency that scales with sequence length and limits the deployment of low-latency or large-scale applications (Gu et al., 2018; Chen et al., 2023; Fu et al., 2024).

Non-autoregressive (NAR) generative models based on diffusion and flow matching have recently emerged as a powerful paradigm for sequence modeling (Austin et al., 2021; Li et al., 2022; Gat et al., 2024; Shaul et al., 2024). These methods enable parallel generation across sequence positions and expose a natural accuracy-efficiency trade-off controlled by the number of inference steps. In particular, Discrete Flow Matching (DFM) (Gat et al., 2024; Campbell et al., 2024) provides a simulation-free framework for training discrete generative models and has shown competitive performance in text domains.

Non-autoregressive approaches for ASR, most notably Connectionist Temporal Classification (CTC) (Graves et al., 2006; Graves & Jaitly, 2014), have been widely adopted, yet remain outside the leading paradigm in state-of-the-art systems. Generative NAR formulations, including recent diffusion and flow-based models, are still emerging, with only limited empirical studies to date (Baas et al., 2022; Yeh et al., 2024; Kwon et al., 2025). As a result, the design space for generative NAR ASR is far less developed than that of AR systems, underscoring the need for principled methods that balance efficiency and accuracy.

Moreover, most applications of DFM have relied on simple path constructions, typically defined as a two-way mixture between a noise-like source distribution (e.g., uniform or masked tokens) and the ground-truth target sequence. This path design means that the model only learns transitions from pure noise to the exact target. While this mismatch may already hinder text generation, it is particularly problematic for ASR: during inference, the model will traverse *acoustically plausible but imperfect* intermediate states, including substitutions, insertions, and deletions (Havasi et al., 2025), that are consistent with the input audio but differ from the ground-truth transcription. The resulting

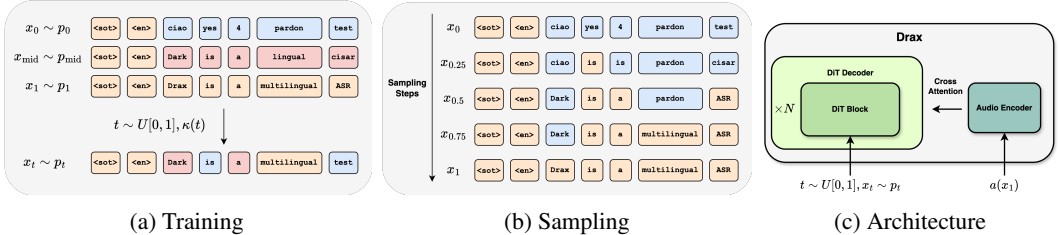

Figure 1: The *Drax* framework: (a) During training, our probability path involves a mixture of three components: a source uniform distribution, the target data distribution, and an audio conditioned distribution. (b) At inference, generation starts from noise tokens and iteratively follows the learned flow to the target sequence, passing through plausible intermediate hypotheses. (c) Drax combines an audio encoder with a DiT-based decoder.

discrepancy between training and inference occupancies resembles the train-sampling mismatch induced by teacher forcing in AR models (Bengio et al., 2015; Ranzato et al., 2015), and motivates the need for richer path designs that better align training dynamics with inference conditions.

In this work, we propose Drax, a framework for NAR ASR based on DFM. Our key idea is to augment the probability path with an *audio-conditioned middle distribution*, which serves as a bridge between the noise-like source and the sharp. Figure 1 illustrates our approach. By exposing the decoder to acoustically consistent but imperfect hypotheses, this tri-mixture path mitigates the domain gap between training and inference. Theoretically, we provide a generalization analysis that relates the risk gap between training and inference to divergences between their respective occupancies. This insight motivates the introduction of an intermediate audio-conditioned distribution and our path design choice. Our experiments demonstrate that the proposed approach achieves competitive accuracy with state-of-the-art ASR baselines while offering favorable runtime-accuracy trade-offs. We further show that Drax benefits from candidate scoring strategies and integrates naturally with speculative decoding, highlighting its potential as an efficient and flexible NAR framework for speech recognition.

Our contributions are as follows: (i) We introduce Drax, a novel non-autoregressive framework for ASR based on a tri-mixture probability path with an audio-conditioned middle distribution. (ii) We provide a theoretical analysis showing that generalization in flow-based models is governed by the divergence between training and inference occupancies, motivating our design. (iii) Through extensive experiments, we demonstrate that Drax improves recognition accuracy over standard DFM paths, and achieves favorable accuracy-efficiency trade-offs compared to AR baselines.

## 2 RELATED WORK

**ASR models.** ASR models are typically trained under two approaches. The earlier approach, CTC (Graves et al., 2006; Graves & Jaitly, 2014), aligns input frames to output sequences without frame-level labels and has been used in wav2vec2.0 (Baevski et al., 2020), MMS (Pratap et al., 2024), and HuBERT (Hsu et al., 2021). Despite its success, CTC has notable drawbacks: CTC assumes conditional independence between tokens and struggles with long-range dependencies, motivating autoregressive approaches. Recent ASR foundation models, such as Whisper (Radford et al., 2023), generate tokens sequentially via a Transformer-based encoder-decoder. Audio-LLM models extend this idea by integrating speech encoders with large language models: Qwen-Audio (Chu et al., 2024) and Canary-Qwen (NeMo, 2025) use Conformer or Whisper encoders with Qwen LLM, while Phi-4-Multimodal (Abouelenin et al., 2025) and Gemma (Gemma Team, 2025) adopt Conformer encoders. These models map audio embeddings into the LM token space. In contrast, models such as AudioPaLM (Rubenstein et al., 2023) and SpeechGPT (Zhang et al., 2023) extend the LM tokens space by introducing dedicated audio tokens. Despite these innovations, inference remains slow due to token-by-token AR decoding.

**NAR generative models.** Recently, diffusion and flow matching generative models (Ho et al., 2020; Song et al., 2020; Nichol & Dhariwal, 2021; Lipman et al., 2022; Tong et al., 2023) have

emerged as NAR alternatives to traditional generative approaches, enabling high-quality samples and more stable sampling. The success of these models has inspired extensions to discrete sequences. Discrete diffusion (Austin et al., 2021; Li et al., 2022) and multinomial diffusion (Hoogeboom et al., 2021) adapt continuous corruption processes to categorical data. Discrete flow models (Campbell et al., 2024; Gat et al., 2024) generalize diffusion by defining probability paths over discrete state-spaces via continuous-time Markov chains. While Campbell et al. (2024) focus on multimodal tasks such as protein co-design, Gat et al. (2024) apply the approach to token distributions, improving text generation. Building on this, Shaul et al. (2024) proposes a general discrete flow matching framework using a kinetic-optimal perspective, enhancing generation quality. Discrete diffusion has also been applied in the speech domain: DCTTS (Wu et al., 2024) uses a discrete latent space with contrastive learning to align text and speech, while DiffS2UT (Zhu et al., 2023) performs reverse diffusion on discrete speech units for speech to speech translation.

**Generative NAR ASR.** Research on generative NAR models for ASR remains very limited. To the best of our knowledge, only Transfusion (Baas et al., 2022), FFDM (Yeh et al., 2024) and the concurrent Whisfusion (Kwon et al., 2025), explore this approach. Both Transfusion and FFDM enable parallel decoding through a multinomial diffusion framework, while Whisfusion combines a Whisper encoder with a diffusion-based decoder, reducing the latency typical of autoregressive models. However, these approaches primarily target English speech and lack evaluation on large-scale or multilingual benchmarks, underscoring the need for further research to assess their generalizability.

## 3 METHOD

In this section, we present Drax, our approach for speech recognition with discrete flow matching. Our goal is to generate a text sequence conditioned on an input audio signal by learning a flow on the space of token sequences. We begin by reviewing the preliminaries of DFM and its formulation as a probability path with an associated velocity field. We then describe our extension, which introduces an audio-conditioned middle distribution to address the mismatch between training and inference, followed by details of the model architecture, training objective, and sampling procedure.

### 3.1 PRELIMINARIES

Let $\mathcal{V}$ denote the vocabulary of tokens of size $d = |\mathcal{V}|$. We denote a sequence of tokens of size $L$ by $x = (x^1, \ldots, x^L) \in \mathcal{V}^L$. In Discrete Flow Matching (Gat et al., 2024), our goal is to learn a generative model mapping a source distribution $p(x_0)$ to a target (data) distribution $q(x_1)$. Let $p_t, t \in [0, 1]$ denote a time-dependent probability mass function (PMF) over $\mathcal{V}^L$, taking the form

$$p_t(x) = \sum_{x_0, x_1 \in \mathcal{V}^L} p_t(x|x_0, x_1)\pi(x_0, x_1), \quad p_t(x|x_0, x_1) = \prod_{i=1}^{L} p_t(x^i|x_0, x_1), \tag{1}$$

where $p_t(x^i|x_0, x_1)$ is a time-dependent conditional probability path on $\mathcal{V}$ which satisfies $p_0(x^i|x_0, x_1) = \delta_{x_0^i}(x^i)$ and $p_1(x^i|x_0, x_1) = \delta_{x_1^i}(x^i)$. Here $(x_0, x_1) \sim \pi$ where $\pi$ is the coupling between source and target. We use unconditional coupling $\pi(x_0, x_1) = p(x_0)q(x_1)$. In this work, we consider the common family of mixture conditional probability paths (Gat et al., 2024; Shi et al., 2024; Sahoo et al., 2024), which are given as a convex sum of $m$ conditional probabilities, $w_j$,

$$p_t(x^i \mid x_0, x_1) = \sum_{j=1}^{m} \kappa_j(t) w_j(x^i \mid x_0, x_1), \tag{2}$$

with $\sum_{j=1}^{m} \kappa_j = 1$, and $\kappa_j \geq 0$, referred to as scheduler. Common choices are the masked and uniform sources with $m = 2$ (Shi et al., 2024; Sahoo et al., 2024; Gat et al., 2024). Following Campbell et al. (2024); Gat et al. (2024), we consider a Continuous-Time discrete Markov Chain (CTMC) with $(X_t)_{t \in [0,1]} \in \mathcal{V}^L$, such that $X_t \sim p_t$. A *probability velocity*, $u_t$, is said to generate the probability path $p_t$ if, for all $t \in [0, 1)$,

$$X_{t+h}^i \sim \delta_{X_t^i}(\cdot) + h \cdot u_t^i(\cdot, X_t) + o(h). \tag{3}$$

Campbell et al. (2024); Gat et al. (2024) show that a generating velocity for $p_t$ can be constructed by considering only the conditional probability paths in Eq. 1. Specifically, given conditional velocities

$u_t^i(x^i, z^i \mid x_0, x_1)$ that generate the conditional paths $p_t(x^i \mid x_0, x_1)$, the marginal probability $u_t$ which generates $p_t$ is given by,

$$u_t^i(x^i, z) = \sum_{x_0, x_1 \in \mathcal{V}^L} u_t^i(x^i, z^i \mid x_0, x_1) p_{1|t}^i(x_0, x_1 \mid z), \tag{4}$$

where $p_{1|t}$ is the posterior probability of $x_0, x_1$ conditioned on the current state $z$. Frequently, training is done using the cross-entropy loss (Gat et al., 2024; Campbell et al., 2024),

$$\mathcal{L}_{\text{CDFM}}(\theta) = -\mathbb{E}_{t,(x_0,x_1),x_t} \sum_{i=1}^{L} \log p_{1|t}^{i,\theta}(x_1^i \mid x_t) \tag{5}$$

## 3.2 Speech Recognition with DFM

We consider the problem of generating a text sequence (tokens), $x_1(a)$, conditioned on input audio $a$. To construct our path, we first consider the simple mixture with $p_0$ a uniform distribution, where the conditional probability path is given by

$$p_t(x^i \mid x_0, x_1) = \kappa_0(t)\delta_{x_0}(x^i) + \kappa_1(t)\delta_{x_1}(x^i). \tag{6}$$

While such a two-component path is simple and effective, it suffers from a fundamental limitation in the ASR setting. During training, the model only observes transitions between pure noise and the ground-truth sequence. At inference time, however, the generative process will traverse states corresponding to *acoustically plausible but imperfect* token sequences. These states may differ from the ground-truth by substitutions, insertions, or deletions that are consistent with the input audio (Havasi et al., 2025). This discrepancy between the training dynamics and the inference dynamics is analogous to teacher forcing in autoregressive models, where the model is exposed only to the true prefixes during training, but must rely on its own generated history during decoding (Williams & Zipser, 1989; Bengio et al., 2015).

**Audio-Bridged Probability Path.** To mitigate this domain gap, we introduce a *middle distribution* $p_{\text{mid}}(\cdot \mid a)$ that is conditioned on the acoustic signal. This distribution reflects sequences that are acoustically probable but may still deviate from the correct transcription, thereby bridging the gap between the uniform source distribution and the sharp ground-truth target distribution. Formally, we define a three-way mixture path of the form

$$p_t(x^i \mid x_0, x_1, a) = \kappa_0(t)\delta_{x_0}(x^i) + \kappa_{\text{mid}}(t)p_{\text{mid}}(x^i \mid a) + \kappa_1(t)\delta_{x_1}(x^i), \tag{7}$$

where $(\kappa_j)_{j \in [0,1,\text{mid}]}$ are smooth mixing schedules that control the interpolation between the source, middle, and target distributions, with $\kappa_0(0) = 1, \kappa_1(1) = 1$ and $\sum \kappa_j(t) = 1, \forall t \in [0,1]$. In practice, we set $\kappa_{\text{mid}}(t)$ to concentrate its transition around $t = 0.5$, ensuring that the middle distribution dominates near the midpoint of the trajectory, see Appendix C.1. By explicitly incorporating $p_{\text{mid}}$, the model is encouraged to learn trajectories that are consistent with both the acoustic signal and plausible intermediate hypotheses (see Figure 1a).

**Model Architecture.** We adopt an encoder-decoder architecture, the dominant paradigm of modern speech recognition models (Radford et al., 2023; Chu et al., 2024). As our audio encoder, we use a pre-trained Whisper encoder (Radford et al., 2023), $E(a) = \varphi_a$. Our decoder uses the DiT architecture (Peebles & Xie, 2023), parametrized by $\theta$, with cross-attention layers to the audio representation in each transformer block, see Figure 1c. The middle distribution $p_{\text{mid}}(\cdot \mid a)$ is parameterized by an auxiliary network $r_\psi$ that takes the encoder representation $\varphi_a$ as input and outputs a per-token categorical distribution.

**Training.** During training, we draw differentiable samples $x_t$ from $p_t$ using the Gumbel-Softmax reparameterization (Maddison et al., 2016; Jang et al., 2016), which allows gradients to propagate into the middle distribution parameters $\psi$. We train the decoder parameters $\theta$ using the cross-entropy loss as in Eq. 5. The middle distribution is trained jointly with the decoder using a combined loss, consists of the standard conditional DFM loss (Eq. 5) and additional auxiliary cross-entropy loss directly on the middle distribution logits,

$$\mathcal{L}_{\text{mid}}(\psi) = -\mathbb{E}_{(a,x_1)} \sum_{i=1}^{L} \log p_{\text{mid}}^{i,\psi}(x_1^i \mid a). \tag{8}$$

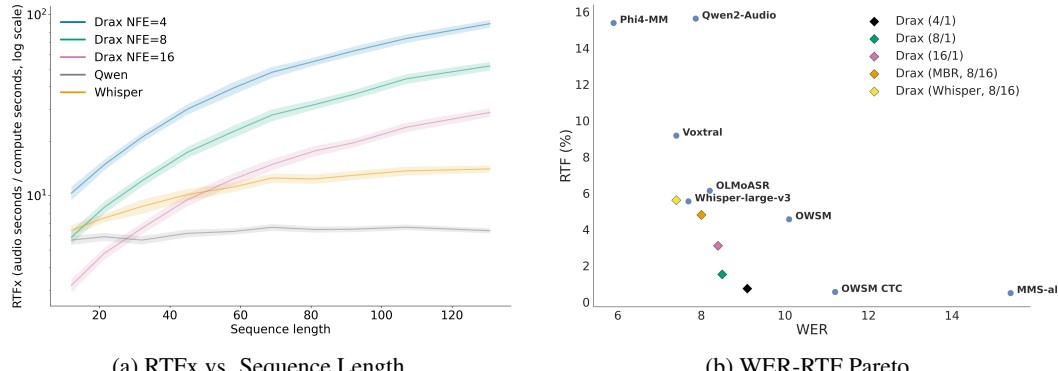

(a) RTFx vs. Sequence Length.  (b) WER-RTF Pareto.

Figure 2: *Accuracy-efficiency trade-off*: (a) The RTFx as a function of sequence length. (b) The Pareto front of the WER and RTF (%) (i.e., 100/RTFx). The Drax varients provide favorable accuracy-efficiency trade-off with better control over the trade-off point.

The final objective is given by

$$\mathcal{L}(\theta, \psi) = \mathcal{L}_{\mathrm{CDFM}}(\theta, \psi) + \mathcal{L}_{\mathrm{mid}}(\psi). \tag{9}$$

**Sampling.**  Sampling proceeds by integrating the marginal velocity field corresponding to the probability path $p_t$. This is done in parallel for each position of the current $X_t$,

$$X_{t+h}^i \sim \delta_{X_t^i}(\cdot) + h u_t^i(\cdot, X_t). \tag{10}$$

In our tri-mixture setup, the marginal velocity is given by,

$$u_t^i(x^i, z) = \alpha_1(t) p_{1|t}^i(x^i \mid z) + \alpha_{\mathrm{mid}}(t) p_{\mathrm{mid}}(x^i) + \beta(t) \delta_z(x^i), \tag{11}$$

where $\alpha, \beta$ depend on the scheduler $\kappa$, see Appendix B. However, in our construction, the middle distribution $p_{\mathrm{mid}}(\cdot \mid a)$ is introduced solely to enrich the training dynamics and expose the decoder to acoustically plausible intermediate states. Thus, at test time we are interested in generating directly from the model without relying on the auxiliary component $p_{\mathrm{mid}}$. Concretely, we therefore set $\alpha_{\mathrm{mid}} \equiv 0$ and sample using the same procedure as in the two-way mixture case (Gat et al., 2024), see Figure 1b. In practice, we use the efficient sampling procedure from Shaul et al. (2024). In Appendix D.6 we provide results for sampling with and without the $p_{\mathrm{mid}}$ component.

**Candidate scoring strategies.**  During sampling, discrete flow matching defines a stochastic generative process. This stochasticity can be exploited at inference time: By sampling multiple candidate transcriptions in parallel and selecting the best one based on a score function, we can enhance both robustness and accuracy. We consider four approaches: (i) First, a simple strategy is to draw several samples and return the most frequent transcription, effectively taking the *mode* over the candidate set. (ii) Second, we consider *minimum Bayes risk* (MBR) decoding. Here, the final prediction is chosen as the candidate with minimum expected word (or character) error rate w.r.t the sampled candidate set, $\mathcal{C}$. Formally,

$$\hat{x}_1 = \arg\min_{x \in \mathcal{C}} \frac{1}{|\mathcal{C}|} \sum_{y \in \mathcal{C}} \mathrm{WER}(x, y). \tag{12}$$

This procedure follows the classical MBR principle (Goel & Byrne, 2000; Kumar & Byrne, 2004; Shen et al., 2016), but leverages the inherent stochasticity of DFM to approximate the posterior expectation through diverse samples. (iii) Third, we rescore candidates using an external model such as Whisper, selecting the sequence with the highest log likelihood under the model. Importantly, this can be done efficiently with a single forward pass in the decoder (Udagawa et al., 2022; Huang et al., 2024). (iv) Finally, following Shaul et al. (2024), the model itself can provide a likelihood-based score by estimating the evidence lower bound (ELBO) along the sampled trajectory, which allows internal ranking without the need for an external scorer.

# 4 THEORETICAL ANALYSIS: OCCUPANCY-BASED BOUND

In this section, we establish a generalization bound for DFM, controlled by the discrepancy between training and inference occupancies, measured in total variation (TV). The analysis highlights the role of the probability path design: As our velocity field is not exact, our sampling trajectory will diverge from the desired path $p_t$. This can worsen the performance of our velocity vector due to covariant shift, thus compounding the error and further widening the gap between our path and the desired trajectory. By reducing this discrepancy, a well-chosen path, such as the audio-conditioned mixture path, can shrink the TV gap, tighten the bound, and improve generalization. Complete and extended formulations of the theoretical results, together with full proofs, are provided in the Appendix A.

**Notations.** Let $t \sim \lambda(t) = \text{Unif}[0, 1]$ and denote $\mathcal{S} = \mathcal{V}^L$ the finite state space. For each $t$, $p_t$ and $q_t$ are probability distributions on $\mathcal{S}$. We define the *occupancies*, using the marginal path distributions:

$$\mu_{\mathcal{D}}(t, x_t) := \lambda(t)\, p_t(x_t), \quad \mu_{\text{gen}}(t, x_t) := \lambda(t)\, q_t(x_t),$$

where,

$$p_t(x_t) = \sum_{x_0, x_1} \pi(x_0, x_1)\, p_t(x_t \mid x_0, x_1), \quad q_t(x_t) = \sum_{x_0} p_0(x_0)\, q_t(x_t \mid x_0).$$

The (target) probability velocity field is written as $u_t(x, z)$, which represents the instantaneous probability flow from $x$ to $z$ that governs the evolution of $p_t$. The learned model velocity is denoted $u_t^\theta(x, z)$, where $\theta$ are the model parameters that governs the evolution of $q_t$.

**Claim 1** (TV stability of path marginals)**.** *Let $\mathcal{S}$ be a finite state space, and let $p_t, q_t \in \Delta(\mathcal{S})$ evolve according to*

$$\dot{p}_t + \text{div}(p_t u_t) = 0, \qquad \dot{q}_t + \text{div}(q_t u_t^\theta) = 0,$$

*Assume $p_0 = q_0$, and define the velocity error $\Delta_t := u_t^\theta - u_t$. Then, for every $s \in [0, 1]$,*

$$\|q_s - p_s\|_{\text{TV}} \leq \int_0^s \mathbb{E}_{x \sim q_t}\Big[\sum_{z \neq x} |\Delta_t(x, z)|\Big] \mathrm{d}t,$$

**Corollary 1** (Instantaneous TV growth)**.** *For almost every $t \in [0, 1]$:*

$$\frac{d}{dt}\|q_t - p_t\|_{\text{TV}} \leq \mathbb{E}_{x \sim p_t}\Big[\sum_{z \neq x} |\Delta_t(x, z)|\Big] + \left\|\sum_{z \neq x} |\Delta_t(x, z)|\right\|_{\infty} \cdot \|q_t - p_t\|_{\text{TV}}$$

The first term represents the *intrinsic model error*, measured under the target distribution $p_t$. The second term captures the additional contribution arising from the mismatch between $q_t$ and $p_t$, i.e., the *domain gap*. Since the velocity field is trained on samples drawn from $p_t$ but applied at inference time to samples from $q_t$, even a moderate error $\Delta_t$ can be amplified by discrepancies between the two distributions, accelerating their divergence over time. We note, however, that this result provides only an upper bound: while it is consistent with our empirical observations, a deeper investigation is required to fully understand the connection.

We can further generalize the discrepancy between our sampling distribution and the target distribution beyond the TV distance in the following theorem:

**Theorem 1** (Generalization bound via occupancy TV)**.** *Assume the instantaneous loss is bounded, $0 \leq \ell_\theta \leq B$. Then*

$$\mathbb{E}_{\mu_{\text{gen}}}[\ell_\theta] \leq \mathbb{E}_{\mu_{\mathcal{D}}}[\ell_\theta] + B\,\big\|\mu_{\text{gen}} - \mu_{\mathcal{D}}\big\|_{\text{TV}} \tag{13}$$

$$\leq \mathbb{E}_{\mu_{\mathcal{D}}}[\ell_\theta] + B \int_0^1 (1 - t)\, \mathbb{E}_{x \sim q_t}\Big[\sum_{z \neq x} |\Delta_t(x, z)|\Big] \mathrm{d}t. \tag{14}$$

|  | LS Clean | LS Other | AMI | Earnings 22 | VoxPopuli | Tedlium | Average | Params (B) | RTFx↑ |
|---|---|---|---|---|---|---|---|---|---|
|  | | | | WER↓ | | | | | |
| Phi4-multimodal | **1.7** | **3.8** | **11.1** | **10.1** | **6.0** | **2.9** | **5.9** | 4.8 | 6.5 |
| Qwen2-Audio | **1.7** | 4.0 | 15.2 | 15.1 | 7.1 | 4.0 | 7.8 | 8.4 | 6.4 |
| Voxtral | 2.0 | 4.3 | 16.8 | 10.6 | 7.1 | 3.6 | 7.4 | 4.7 | 10.9 |
| MMS-all | 3.8 | 8.3 | 36.4 | 24.6 | 9.6 | 10.1 | 15.4 | 1.0 | **201.2** |
| OWSM CTC | 3.3 | 6.6 | 25.5 | 18.6 | 8.4 | 4.9 | 11.2 | 1.0 | 178.3 |
| Whisper-large-v3 | 2.0 | 3.9 | 16.2 | 11.1 | 8.8 | 3.9 | 7.6 | 1.5 | 18.0 |
| OLMoASR-large.en-v2 | 2.7 | 5.6 | 16.8 | 11.9 | 8.0 | 4.2 | 8.2 | 1.5 | 16.3 |
| OWSM | 2.4 | 5.3 | 23.9 | 15.8 | 8.3 | 5.0 | 10.1 | 1.0 | 21.9 |
| TransFusion | 6.7 | 8.8 | – | – | – | – | – | 0.2 | – |
| Whisfusion | 8.3 | 17.0 | – | – | – | – | – | 0.3 | – |
| FDDM | 4.0 | 7.2 | – | – | – | – | – | 0.6 | – |
| DFM | 3.4 | 7.0 | 30.8 | 17.6 | 10.5 | 6.9 | 12.7 | 1.2 | 32.2 |
| Drax | 2.6 | 5.7 | 13.9 | 15.2 | 8.6 | 4.8 | 8.4 | 1.2 | 32.2 |
| Drax (MBR, 8/16) | 2.6 | 5.3 | 13.6 | 14.6 | 8.0 | 4.1 | 8.0 | 1.2 | 20.8 |
| Drax (Whisper, 8/16) | 2.2 | 4.7 | 12.7 | 13.7 | 7.4 | 3.7 | 7.4 | 2.1 | 17.8 |

Table 1: Results for English datasets from the HF benchmark (Srivastav et al., 2023). Drax provide control over the accuracy-efficiency trade-off, and achieve on-par results with SoTA methods.

## 5 EXPERIMENTS

**Baselines.** We evaluate Drax against several recent methods. *Whisper* (Radford et al., 2023) (`large-v3`) is an encoder-decoder multilingual model for speech recognition and translation, trained on 5M hours of weakly supervised speech-text pairs. *Qwen2-Audio* (Chu et al., 2024) extends the Qwen2 language model (Yang et al., 2024) with an audio encoder. *Phi4-multimodal, an LLM-based model with vision, language and audio capabilities. Voxtral* (Liu et al., 2025) is built on the Mistral LLM backbone and incorporates a dedicated speech encoder. *OLMoASR* (Ngo et al., 2025) (`large.en-v2`) is trained on up to three million curated hours of English-only speech data. *OWSM* (Peng et al., 2024) is a fully open multilingual speech model, with an AR and CTC based varients. *MMS* (Pratap et al., 2024) is a large-scale multilingual speech model trained with a CTC objective, supporting speech recognition in over 100 languages. In addition, *TransFusion* (Baas et al., 2022) and *FDDM* (Yeh et al., 2024) use multinomial diffusion, while *Whisfusion* (Kwon et al., 2025) employs a diffusion-based Transformer for parallel ASR decoding. For Drax, we denote e.g., Drax(MBR, 8/16) with the convention (scoring method, NFE/Candidate set size). If not otherwise stated, we use 16 NFE and a generate a single transcription, i.e. Drax (16/1). We also report a *DFM baseline, which is a Drax variant trained with the standard mixture path.* Together, these baselines cover large-scale encoder-decoder models, LLM-based ASRs, and CTC and diffusion based models.

**Evaluation Metrics.** We report performance using both recognition accuracy and computational efficiency. Accuracy is measured using *word error rate* (WER) for languages with explicit word segmentation, and *character error rate* (CER) for languages such as Chinese or Japanese, where text is naturally represented at the character level. For efficiency, we measure runtime using the *real-time factor* (RTF), defined as the ratio between compute time and input audio duration. We also report its reciprocal RTFx = (audio seconds) / (compute seconds). A value RTFx > 1 indicates faster-than-real-time decoding. This metric allows for a direct comparison of accuracy-efficiency trade-offs across different model families.

**Training details.** We train two variants of our method, on top of the Whisper (`large-v3`) encoder: Drax, which is composed of 16 decoder blocks with 20 attention heads and 1280 hidden dimension, and Drax-flash, a smaller variant with a similar configuration but only 4 decoder layers. The DiT (Peebles & Xie, 2023) decoders of Drax and Drax-flash consists of 580M and 250M parameters, respectively. We adapt the same tokenizer as in Radford et al. (2023). The audio conditioned distribution $p_{\text{mid}}$ contains a single transformer block together with a projection layer, to output the per-position logits over the vocabulary, with a total of 28M parameters. The models are trained with a mix of 8 languages (English, Spanish, German, French, Italian, Portuguese, Chinese, and Japanese) with a total of 15K hours. See Appendix C.2 for more details.

### 5.1 MULTILINGUAL SPEECH RECOGNITION

We begin by evaluating the multilingual performance of Drax against strong encoder-decoder and CTC baselines on a broad suite of public ASR benchmarks (see Appendix C.2). The suite spans

| | WER↓ | | | | | | | | | | | | | | CER↓ | | | |
| | DE | | | ES | | | FR | | | IT | | | PT | | JA | | ZH | |
| | MLS | CV | Vox. | MLS | CV | Vox. | MLS | CV | Vox. | MLS | CV | Vox. | MLS | CV | CV | Reazon | CV | AISHELL |
|---|---|---|---|---|---|---|---|---|---|---|---|---|---|---|---|---|---|---|
| Qwen2-Audio | 8.1 | 7.5 | 12.5 | 5.4 | 5.7 | 9.6 | 5.7 | 9.5 | 15.2 | 12.5 | 6.7 | 19.2 | 11.6 | 9.1 | 15.2 | 50.7 | **7.0** | **1.5** |
| Voxtral | 7.6 | 6.2 | **11.0** | 5.1 | **4.7** | **8.6** | 5.4 | **8.9** | 14.7 | 11.2 | **5.5** | 16.7 | **6.6** | 6.2 | – | – | – | – |
| MMS-all | 8.6 | 12.3 | 16.1 | 5.7 | 9.9 | 10.9 | 8.7 | 16.0 | 18.3 | 11.0 | 9.90 | 19.8 | 15.8 | 11.7 | 31.0 | 49.2 | 25.6 | 31.2 |
| OWSM CTC | 11.8 | 11.4 | 16.4 | 10.3 | 11.6 | 14.9 | 12.9 | 15.4 | 21.1 | 22.1 | 15.2 | 25.8 | 23.5 | 19.6 | **12.2** | **10.4** | 13.2 | 6.3 |
| Whisper-large-v3 | **5.5** | **6.0** | 13.1 | **3.9** | 5.0 | 10.5 | **4.7** | 11.3 | 15.0 | **9.2** | 5.8 | 28.5 | 7.1 | **5.7** | **12.2** | 19.1 | 16.1 | 8.8 |
| OWSM | 11.0 | 10.2 | 16.4 | 9.0 | 10.6 | 15.5 | 12.1 | 15.0 | 21.3 | 20.2 | 13.8 | 35.1 | 22.3 | 20.3 | 14.9 | 14.6 | 14.5 | 6.4 |
| Drax | 7.7 | 9.1 | 11.9 | 5.4 | 6.8 | 10.6 | 7.1 | 12.0 | 17.0 | 12.5 | 8.5 | 18.0 | 13.6 | 11.5 | 14.1 | 13.4 | 18.0 | 7.8 |
| Drax (MBR, 8/16) | 7.3 | 8.5 | 12.5 | 4.9 | 6.2 | 10.2 | 6.4 | 11.2 | 11.8 | 11.0 | 7.7 | 16.5 | 11.9 | 10.6 | 13.2 | 12.5 | 16.5 | 8.7 |
| Drax (Whisper, 8/16) | 6.5 | 7.2 | 11.4 | 4.3 | 5.3 | 9.0 | 5.8 | 10.1 | **10.6** | 10.3 | 6.5 | **16.0** | 10.8 | 8.6 | 12.7 | 12.2 | 15.3 | 6.7 |

Table 2: *Multilingual evaluation*: Results on the MLS, CommonVoice-13, VoxPopuli, Reazon-Speech and AISHELL datasets.

multiple language families and scripts, mixes read and spontaneous speech, and covers diverse domains (conversational, technical, formal) under both clean and noisy acoustic conditions. Together, these benchmarks constitute a comprehensive test of robustness to domain shift and cross-lingual generalization. The results are presented in Tables 1 and 2. Bold indicates the best performance, underline indicates the second-best. Across a wide range of English and multilingual benchmarks, Drax is on-par or surpasses strong ASR baselines. The model's performance is consistent across domains and languages highlighting its robustness. Variants that use MBR or Whisper-guided scoring provide additional gains with minimal impact on throughput. Overall, the results validate Drax as a competitive and efficient approach to multilingual ASR. Extended results with different Drax variants are presented in Appendix D.3.

## 5.2 Accuracy-Efficiency Trade-off

A key efficiency advantage of NAR approaches such as flow-matching and diffusion-based models over AR decoders is that sampling is inherently parallel across sequence positions (Austin et al., 2021; Li et al., 2022; Gat et al., 2024). While AR models like Whisper and Qwen2-Audio must decode tokens sequentially, causing latency to scale with output length (Radford et al., 2023; Chu et al., 2024), Drax requires only a fixed number of function evaluations (NFE). This design not only reduces dependence on sequence length but also enables explicit control over the accuracy-efficiency trade-off: increasing NFE improves WER, while smaller NFE yields faster decoding. Figure 2a demonstrates the scaling advantage with respect to utterance length, and Figure 2b illustrates the Pareto frontier of WER versus runtime measured by RTF (1/RTFx).

Beyond NFE, accuracy can also be traded for efficiency through candidate generation and scoring. The DFM framework naturally supports sampling multiple candidates for a given audio input, and we evaluate several scoring strategies described in Section 3.2. Figures 6 and 10 in the supplementary show the effect of candidate set size and temperature on WER. We observe that ELBO-based scoring (Shaul et al., 2024) is unstable, while MBR consistently achieves strong results, comparable with Whisper scoring. Whisper scoring itself is efficient since it requires only a single decoder forward pass with the candidate batch. Smaller temperature values reduce diversity but improve average candidate quality. Table 5 reports results under varying NFE and candidate set sizes, together with RTFx. We select 8 NFE and 16 candidates as a good trade-off between accuracy and efficiency. Finally, across all settings we observe a notable gap between the best scoring strategy and the oracle (minimum candidates WER), highlighting future opportunities for improving candidate selection.

## 5.3 Speculative Decoding

We evaluate Drax under a speculative decoding scheme, where a fast generator proposes continuations that are verified by a stronger model to reduce wall-clock latency without sacrificing accuracy (Leviathan et al., 2023). Importantly, a non-autoregressive drafter avoids the per-token dependency of autoregressive hypothesis generation, enabling parallel block proposals. These proposals cover multi-token continuations in one (or a few) forward passes, substantially reducing runtime overhead without compromising prediction accuracy (Chen et al., 2024; Wen et al., 2024). In our setup, the Drax model serves as the draft model, and Whisper acts as the target model. During verification, a draft token is accepted only if it matches the top-1 prediction of the target model.

| | DE | | ES | | FR | | IT | | PT | | LS-clean | | LS-other | |
|---|---|---|---|---|---|---|---|---|---|---|---|---|---|---|
| | #Matches | RTFx | #Matches | RTFx | #Matches | RTFx | #Matches | RTFx | #Matches | RTFx | #Matches | RTFx | #Matches | RTFx |
| Whisper large-v3 | – | 18.36 | – | 19.01 | – | 17.47 | – | 18.85 | – | 18.60 | – | 16.02 | – | 15.64 |
| Whisper-turbo (10) | 2.31 | 15.90 | 4.73 | 26.02 | 2.83 | 17.12 | 3.00 | 19.38 | 3.66 | 22.08 | 5.07 | 22.50 | 4.41 | 20.18 |
| Whisper-turbo (5) | 1.94 | 18.80 | 3.16 | 26.12 | 2.21 | 19.32 | 2.36 | 21.76 | 2.70 | 23.61 | 3.20 | 21.67 | 2.92 | 19.96 |
| Drax-flash | **10.57** | **38.54** | **11.27** | **42.17** | **8.25** | **31.88** | **7.03** | **31.04** | **5.01** | **25.15** | **6.81** | **24.59** | **5.15** | **20.39** |

Table 3: *Speculative decoding*: Using Drax-flash for generating hypothesis for a target model (Whisper large-v3). Drax-flash outperforms Whisper turbo in both number of matched tokens and RTFx.

We compare Drax and Whisper-Turbo as draft models. Whisper-Turbo is run in two configurations that speculate 5 or 10 tokens per step. For Drax we use NFE $= 2$ and $\tau = 0.01$. We evaluate the models on MLS, LS-clean, and LS-other and report per-language results. Specifically, we report both RTFx and the number of matched tokens; the latter represents the average number of token candidates that the target model approves per step. The results are presented in Table 3. Drax yields substantial speedups over Whisper-Turbo when used as the draft model, especially on non-English languages. These results highlight an important application of Drax as a NAR ASR.

## 5.4 Training Path Design

We conduct an experiment to study the effect of different probability paths on model generalization. We train a compact DiT decoder (12 layers, hidden size 768, 205M parameters) with a frozen Whisper-small encoder (88M parameters) under four path configurations: (i) a uniform source baseline, (ii) a uniform source with uniform middle, (iii) a uniform source with audio-conditioned middle, and, (iv) an audio-conditioned source. All models are trained for 100K steps and evaluated with 8 NFEs. Figure 3 reports the generalization word error rate over the training trajectory. We observe that the uniform source-audio middle path achieves the lowest WER throughout training, significantly outperforming both the uni-

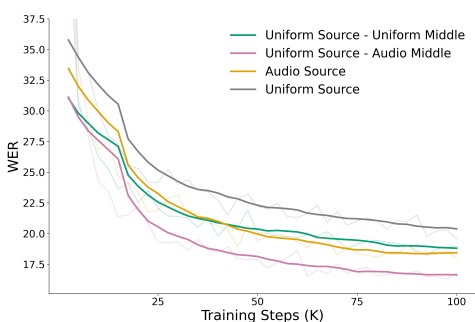

Figure 3: *Training path design*. Comparison of training curves under different paths.

form and audio-conditioned source only paths. Introducing a middle distribution consistently improves generalization compared to direct source-target paths, with the audio-conditioned middle providing the largest gains. In addition, the results in Table 1 show a similar pattern: Drax significantly outperforms the DFM baseline, corresponds to the unifrom source variant, with average WER of 8.4 vs. 12.7. These results validate our hypothesis that exposing the model to acoustically plausible intermediate states during training improves robustness and reduces errors.

## 6 Discussion

**Limitations.** While Drax demonstrates strong recognition accuracy and favorable efficiency trade-offs compared to AR and existing NAR ASR systems, several limitations remain. First, our experiments are conducted on a curated set of public multilingual datasets; scaling to much larger or more diverse training corpora may reveal additional challenges in robustness and generalization. In addition, although we show that introducing an intermediate distribution improves alignment between training and inference, the design of probability paths for ASR remains largely unexplored. Our choice of an audio-conditioned middle distribution is only one instantiation, and future work should investigate alternative or adaptive path constructions.

**Conclusion.** In this work, we introduced Drax, a discrete flow matching framework for NAR speech recognition that leverages a tri-mixture probability path with an audio-conditioned middle distribution. We provide theoretical analysis which to motivate our path design choice. Empirically, Drax achieves competitive performance with state-of-the-art, large scale ASR models while offering improved runtime efficiency, and it integrates naturally with candidate scoring and speculative decoding strategies. These findings highlight discrete flow matching as a promising foundation for future non-autoregressive ASR research.

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

# A PROOFS

Here, we provide full theoretical derivation and missing proofs for Section 4.

**Setup and notation.** Let $(x_0, x_1) \sim \pi$ be a coupling between source and data (e.g., $\pi(x_0, x_1) = p_0(x_0)\, p_{\text{data}}(x_1)$), and let $t \sim \text{Unif}[0, 1]$. The finite state space is $\mathcal{S} = \mathcal{V}^L$, where $\mathcal{V}$ is the vocabulary of tokens and $L$ the sequence length. For each $t$, $p_t$ and $q_t$ are probability distributions on $\mathcal{S}$. During training, $p_t(\cdot \mid x_0, x_1)$ denotes the designed conditional path; during generation, the model defines $q_t(\cdot \mid x_0)$.

We define the *occupancies*, which can be measured either at the sequence or site level, using the marginal path distributions:

$$\mu_{\mathcal{D}}(t, x_t) := \lambda(t)\, p_t(x_t), \quad \mu_{\text{gen}}(t, x_t) := \lambda(t)\, q_t(x_t),$$

where $\lambda = \text{Unif}[0, 1]$ is the base time measure, and

$$p_t(x_t) = \sum_{x_0, x_1} \pi(x_0, x_1)\, p_t(x_t \mid x_0, x_1),$$

$$q_t(x_t) = \sum_{x_0} p_0(x_0)\, q_t(x_t \mid x_0).$$

On the finite state space $\mathcal{S}$, let $x, z \in \mathcal{S}$ denote states. The (target) probability velocity field is written as $u_t(x, z) \geq 0$ for $z \neq x$, which represents the instantaneous probability flow from $x$ to $z$ and governs the evolution of $p_t$. The learned model velocity is denoted $u_t^\theta(x, z)$, where $\theta$ are the model parameters that governs the evolution of $q_t$.

We define the *velocity error* by

$$\Delta_t(x, z) := u_t^\theta(x, z) - u_t(x, z),$$

The discrete divergence operator acts on fluxes $v(x, z)$ between states, and for each state $x$ returns the imbalance between incoming and outgoing flow:

$$\text{div}_x(v) = \sum_{z \in \mathcal{S}} \big[ v(z, x) - v(x, z) \big].$$

In other words, $\text{div}_x(v)$ equals the total inflow into state $x$ minus the total outflow from $x$, expressing the conservation law that governs how probability mass moves across states.

The evolution of the distributions is governed by the continuity equations, also known in the Markov chain literature as the master equations:

$$\dot{p}_t + \text{div}(p_t u_t) = 0, \qquad \dot{q}_t + \text{div}(q_t u_t^\theta) = 0.$$

Assume the instantaneous loss is bounded, $0 \leq \ell_\theta \leq B$, and denote the training and generation risks by

$$R_{\mathcal{D}}(\theta) = \mathbb{E}_{\mu_{\mathcal{D}}}[\ell_\theta], \qquad R_{\text{gen}}(\theta) = \mathbb{E}_{\mu_{\text{gen}}}[\ell_\theta].$$

**Claim 1** [TV stability of path marginals] Let $\mathcal{S}$ be a finite state space, and let $p_t, q_t \in \Delta(\mathcal{S})$ evolve according to

$$\dot{p}_t + \text{div}(p_t u_t) = 0, \qquad \dot{q}_t + \text{div}(q_t u_t^\theta) = 0,$$

where $u_t, u_t^\theta : \mathcal{S} \times \mathcal{S} \to \mathbb{R}$ are velocity fields satisfying $u_t(x, z) \geq 0$, $u_t^\theta(x, z) \geq 0$ for all $z \neq x$, and

$$u_t(x, x) = -\sum_{z \neq x} u_t(x, z), \qquad u_t^\theta(x, x) = -\sum_{z \neq x} u_t^\theta(x, z).$$

Assume $p_0 = q_0$, and define the velocity error $\Delta_t := u_t^\theta - u_t$. Then, for every $s \in [0, 1]$,

$$\|q_s - p_s\|_{\text{TV}} \leq \int_0^s \mathbb{E}_{x \sim q_t} \Big[ \sum_{z \neq x} |\Delta_t(x, z)| \Big]\, dt$$

*Proof.* This proof follows the overall strategy of Huang et al., who establish the result in the continuous setting. Here, we adapt and extend their argument to the discrete case for completeness.

Let $r_t := q_t - p_t$. Subtracting the continuity equations

$$\dot{q}_t + \operatorname{div}(q_t u_t^\theta) = 0, \qquad \dot{p}_t + \operatorname{div}(p_t u_t) = 0,$$

yields

$$\dot{r}_t = -\operatorname{div}(q_t u_t^\theta - p_t u_t) = -\operatorname{div}(r_t u_t) - \operatorname{div}(q_t(u_t^\theta - u_t)).$$

With $\Delta_t := u_t^\theta - u_t$, this becomes

$$\dot{r}_t = -\operatorname{div}(r_t u_t) - \operatorname{div}(q_t \Delta_t),$$

with initial condition $r_0 = 0$ since $p_0 = q_0$. Note that $r_t$ is not a probability distribution but a signed measure satisfying $\sum_x r_t(x) = 0$.

The homogeneous system

$$\dot{h}_t = -\operatorname{div}(h_t u_t)$$

induces a time-inhomogeneous Markov evolution operator $S_{t \to s}$, defined as the linear map that propagates a distribution $h_t$ at time $t$ to its state at time $s$: $h_s = S_{t \to s} h_t$ (van, 2011). By the variation-of-constants (Duhamel) (Bers et al., 1964) formula for linear ODEs with bounded generators, the solution of the inhomogeneous system is

$$r_s = S_{0 \to s} r_0 - \int_0^s S_{t \to s} \operatorname{div}(q_t \Delta_t) \, dt.$$

Since $r_0 = 0$, this simplifies to

$$r_s = -\int_0^s S_{t \to s} \operatorname{div}(q_t \Delta_t) \, dt.$$

According to Theorem 3.33 (Dynkin) in van (2011), the evolution operator $S_{t \to s}$ is contractive, hence:

$$\|S_{t \to s} \phi\|_{\mathrm{TV}} \le \|\phi\|_{\mathrm{TV}}.$$

Applying this with $\phi = \operatorname{div}(q_t \Delta_t)$ inside the Duhamel representation,

$$r_s = -\int_0^s S_{t \to s} \operatorname{div}(q_t \Delta_t) \, dt,$$

we obtain

$$\|r_s\|_{\mathrm{TV}} = \left\| \int_0^s S_{t \to s} \operatorname{div}(q_t \Delta_t) \, dt \right\|_{\mathrm{TV}}.$$

First, by the triangle inequality for vector-valued integrals (i.e., $\| \int_0^s X_t dt \| \le \int_0^s \|X_t\| dt$)

$$\left\| \int_0^s S_{t \to s} \operatorname{div}(q_t \Delta_t) \, dt \right\|_{\mathrm{TV}} \le \int_0^s \left\| S_{t \to s} \operatorname{div}(q_t \Delta_t) \right\|_{\mathrm{TV}} dt.$$

Second, by TV-contraction of the Markov evolution $S_{t \to s}$,

$$\left\| S_{t \to s} \operatorname{div}(q_t \Delta_t) \right\|_{\mathrm{TV}} \le \left\| \operatorname{div}(q_t \Delta_t) \right\|_{\mathrm{TV}}.$$

Combining the two displays gives

$$\|r_s\|_{\mathrm{TV}} \le \int_0^s \| \operatorname{div}(q_t \Delta_t) \|_{\mathrm{TV}} \, dt. \tag{15}$$

For each $x \in \mathcal{S}$,

$$(\operatorname{div}(q_t \Delta_t))(x) = \sum_{z \neq x} \big( q_t(z) \Delta_t(z, x) - q_t(x) \Delta_t(x, z) \big).$$

Hence:

$$\| \operatorname{div}(q_t \Delta_t) \|_{\mathrm{TV}} = \frac{1}{2} \sum_{x \in \mathcal{S}} \Big| \sum_{z \neq x} (q_t(z) \Delta_t(z, x) - q_t(x) \Delta_t(x, z)) \Big| \leq \sum_x q_t(x) \sum_{z \neq x} |\Delta_t(x, z)|.$$

By dropping the inflow terms and upper bounding with the total outflow, we obtain

$$\| \operatorname{div}(q_t \Delta_t) \|_{\mathrm{TV}} \leq \sum_x q_t(x) \sum_{z \neq x} |\Delta_t(x, z)|.$$

Combining equation 15 with this bound yields

$$\|q_s - p_s\|_{\mathrm{TV}} = \|r_s\|_{\mathrm{TV}} \leq \int_0^s \mathbb{E}_{x \sim q_t} \Big[ \sum_{z \neq x} |\Delta_t(x, z)| \Big] \, dt.$$

In particular, for $s = 1$,

$$\|q_1 - p_1\|_{\mathrm{TV}} \leq \int_0^1 \mathbb{E}_{x \sim q_t} \Big[ \sum_{z \neq x} |\Delta_t(x, z)| \Big] \, dt.$$

This completes the proof. $\qquad\square$

**Corollary 1** [Instantaneous TV growth] For a.e $t \in [0, 1]$,

$$\frac{d}{dt} \|q_t - p_t\|_{\mathrm{TV}} \leq \mathbb{E}_{x \sim q_t} \Big[ \sum_{z \neq x} |\Delta_t(x, z)| \Big].$$

which can be decomposed into two parts:

$$\frac{d}{dt} \|q_t - p_t\|_{\mathrm{TV}} \leq \mathbb{E}_{x \sim p_t} \Big[ \sum_{z \neq x} |\Delta_t(x, z)| \Big] + \Big( \mathbb{E}_{x \sim q_t} \Big[ \sum_{z \neq x} |\Delta_t(x, z)| \Big] - \mathbb{E}_{x \sim p_t} \Big[ \sum_{z \neq x} |\Delta_t(x, z)| \Big] \Big)$$

$$\leq \mathbb{E}_{x \sim p_t} \Big[ \sum_{z \neq x} |\Delta_t(x, z)| \Big] + \Big\| \sum_{z \neq x} |\Delta_t(x, z)| \Big\|_\infty \cdot \|q_t - p_t\|_{\mathrm{TV}}.$$

$$(16)$$

The first term reflects the *intrinsic model error* under $p_t$, while the second term quantifies the extra contribution from the *domain gap* between $q_t$ and $p_t$.

*Proof.* By Claim A, for all $s \in [0, 1]$,

$$\|q_s - p_s\|_{\mathrm{TV}} \leq \int_0^s \mathbb{E}_{x \sim q_t} \Big[ \sum_{z \neq x} |\Delta_t(x, z)| \Big] \, dt.$$

The right-hand side is absolutely continuous in $s$, hence so is $s \mapsto \|q_s - p_s\|_{\mathrm{TV}}$. By the fundamental theorem of calculus for absolutely continuous functions, the derivative exists for a.e. $t$ and satisfies

$$\frac{d}{dt} \|q_t - p_t\|_{\mathrm{TV}} \leq \mathbb{E}_{x \sim q_t} \Big[ \sum_{z \neq x} |\Delta_t(x, z)| \Big]$$

.

$\qquad\square$

**Proposition 1** (From path-marginal TV to occupancy TV). *With*

$$\mu_{\mathcal{D}}(t, x_t) := \lambda(t) \, p_t(x_t), \quad \mu_{\mathrm{gen}}(t, x_t) := \lambda(t) \, q_t(x_t),$$

*where* $\lambda = \mathrm{Unif}[0, 1]$, *we have*

$$\big\| \mu_{\mathrm{gen}} - \mu_{\mathcal{D}} \big\|_{\mathrm{TV}} = \mathbb{E}_{t \sim \mathrm{Unif}[0,1]} \|q_t - p_t\|_{\mathrm{TV}}, \tag{17}$$

$$\leq \int_0^1 (1 - t) \, \mathbb{E}_{x \sim q_t} \Big[ \sum_{z \neq x} |\Delta_t(x, z)| \Big] \, dt. \tag{18}$$

*Proof.* The difference of occupancies is

$$(\mu_{\text{gen}} - \mu_{\mathcal{D}})(t, x) = \lambda(t)\big(q_t(x) - p_t(x)\big).$$

By the variational characterization of total variation on the product space $\mathcal{S} \times [0, 1]$,

$$\|\mu_{\text{gen}} - \mu_{\mathcal{D}}\|_{\text{TV}} = \sup_{\|f\|_\infty \leq 1} \int_0^1 \sum_{x \in \mathcal{S}} f(x, t)\big(q_t(x) - p_t(x)\big)\lambda(t)\, dt.$$

Since $\mathcal{S}$ is finite and $t \mapsto q_t(x) - p_t(x)$ is measurable for each $x$, the selector $g_t(x) = \text{sign}(q_t(x) - p_t(x))$ is measurable in $t$, as the sign map is Borel-measurable (Castaing & Valadier, 2006).

Hence $f(x, t) = g_t(x)$ is an admissible measurable test function on $\mathcal{S} \times [0, 1]$ that attains the inner supremum pointwise in $t$. This justifies exchanging the supremum and the integral, yielding

$$\|\mu_{\text{gen}} - \mu_{\mathcal{D}}\|_{\text{TV}} = \int_0^1 \|q_t - p_t\|_{\text{TV}}\,\lambda(t)\, dt.$$

Equivalently,

$$\|\mu_{\text{gen}} - \mu_{\mathcal{D}}\|_{\text{TV}} = \mathbb{E}_{t \sim \text{Unif}[0,1]}\big[\|q_t - p_t\|_{\text{TV}}\big].$$

By Claim A with $s = t$,

$$\|q_t - p_t\|_{\text{TV}} \leq \int_0^t \|\Delta_\tau\|_{\text{row-}\ell_1}\, d\tau,$$

Taking expectation over $t \sim \text{Unif}[0, 1]$ and setting $\psi(\tau) := \mathbb{E}_{x \sim q_\tau}\Big[\sum_{z \neq x}|\Delta_\tau(x, z)|\Big]$

$$\begin{aligned}
\mathbb{E}_t[\|q_t - p_t\|_{\text{TV}}] &\leq \mathbb{E}_t\left[\int_0^t \psi(\tau)\, d\tau\right] \\
&= \int_0^1 \int_0^t \psi(\tau)\, d\tau\, dt \\
&= \int_0^1 \int_\tau^1 \psi(\tau)\, dt\, d\tau \\
&= \int_0^1 (1 - \tau)\,\psi(\tau)\, d\tau \\
&= \int_0^1 (1 - t)\,\mathbb{E}_{x \sim q_t}\Big[\sum_{z \neq x}|\Delta_t(x, z)|\Big]\, dt.
\end{aligned}$$

Combining this with equation 17, we obtain

$$\|\mu_{\text{gen}} - \mu_{\mathcal{D}}\|_{\text{TV}} = \mathbb{E}_{t \sim \text{Unif}[0,1]}\big[\|q_t - p_t\|_{\text{TV}}\big] \leq \int_0^1 (1 - t)\,\mathbb{E}_{x \sim q_t}\Big[\sum_{z \neq x}|\Delta_t(x, z)|\Big]\, dt$$

which is exactly inequality equation 18.

$$\square$$

**Theorem 1** [DA-style generalization bound via occupancy TV] Assume the instantaneous loss is bounded, $0 \leq \ell_\theta \leq B$. Then

$$R_{\text{gen}}(\theta) \leq R_{\mathcal{D}}(\theta) + B\,\big\|\mu_{\text{gen}} - \mu_{\mathcal{D}}\big\|_{\text{TV}}, \tag{19}$$

$$R_{\text{gen}}(\theta) \leq R_{\mathcal{D}}(\theta) + B\int_0^1 (1 - t)\,\mathbb{E}_{x \sim q_t}\Big[\sum_{z \neq x}|\Delta_t(x, z)|\Big]\, dt. \tag{20}$$

*Proof.* By TV duality, for any bounded $f$ with $\|f\|_\infty \leq B$ and any probability measures $P, Q$,

$$\left| \mathbb{E}_P f - \mathbb{E}_Q f \right| \;\leq\; B \, \|P - Q\|_{\mathrm{TV}}.$$

Apply this with $f = \ell_\theta$, $P = \mu_{\mathrm{gen}}$, $Q = \mu_{\mathcal{D}}$ to obtain equation 19. Then substitute the bound from Proposition 1,

$$\|\mu_{\mathrm{gen}} - \mu_{\mathcal{D}}\|_{\mathrm{TV}} \;=\; \mathbb{E}_{t \sim \mathrm{Unif}[0,1]} \|q_t - p_t\|_{\mathrm{TV}} \;\leq\; \int_0^1 (1-t)\, \mathbb{E}_{x \sim q_t} \left[ \sum_{z \neq x} |\Delta_t(x, z)| \right] dt,$$

which yields equation 20. $\qquad\qquad\qquad\qquad\qquad\qquad\qquad\qquad\qquad\qquad\qquad\qquad\qquad\qquad\square$

## B   TRI-MIXTURE VELOCITIES

For completeness, we provide the form of $u_t(\cdot)$ and $u_t(\cdot \mid x_0, x_1)$ in our tri-mixture path. Using Theorems 2 and 3 in Gat et al. (2024), we have

$$u_t^i(a, z) = \sum_j \alpha_t^{i,j} \hat{w}_i^j(a, z) + \beta_t^i \delta_{a, z_i}, \tag{21}$$

where $\hat{w}$ the posterior of $w$ is defined as,

$$\hat{w}_t^j(a, z) = \sum_{x_0, x_1} w^j(a \mid x_0, x_1) p_t(x_0, x_1 \mid z). \tag{22}$$

Theorem 3 gives the coefficients as $\alpha_t^{i,j} = \dot{\kappa}_t^{i,j} - \kappa_t^{i,j} \dot{\kappa}_t^{i,\ell} / \kappa_t^{i,\ell}$, $\beta_t^i = \dot{\kappa}_t^{i,\ell} / \kappa_t^{i,\ell}$ with $\ell = \arg\min_j \dot{\kappa}_t^{i,j} / \kappa_t^{i,j}$. In our case $w^1(a \mid x_0, x_1) = \delta_{x_1^i}(a), w^{\mathrm{mid}}(a \mid x_0, x_1) = p_{\mathrm{mid}}^i(a), w^0(a \mid x_0, x_1) = \delta_{x_0^i}(a)$ and the marginal posterior $w_t^{i,1}(a, z) = p_{1|t}^i(a|z), w_t^{i,\mathrm{mid}}(a, z) = p_{\mathrm{mid}}^i(a), w_t^{i,0}(a, z) = p_{0|t}^i(a|z)$ since $p_{\mathrm{mid}}$ is independent of the endpoints. Thus, we have,

$$u_t^i(a, z) = \alpha_t^1 p_{1|t}^i(a \mid z) + \alpha_t^{\mathrm{mid}} p_{\mathrm{mid}}^i(a) + \alpha_t^0 p_{0|t}^i(a) + \beta_t \delta_z(a), \tag{23}$$

and the conditional probability velocity is given by,

$$u_t^i(a, z \mid x_0, x_1) = \alpha_t^1 \delta_{x_1^i}(a) + \alpha_t^{\mathrm{mid}} p_{\mathrm{mid}}^i(a) + \alpha_t^0 \delta_{x_0^i}(a) + \beta_t \delta_z(a). \tag{24}$$

Now, our scheduler construction, as provided in Appendix C.1, ensures $\ell \equiv 0$, and so the terms $p_{0|t}^i(a)$ and $\delta_{x_0^i}(a)$ are dropped from $u_t$ and $u_t(\cdot \mid x_0, x_1)$, respectively.

## C   EXPERIMENTAL DETAILS

### C.1   TRI-MIXTURE SCHEDULER

Training with a three-way probability path requires mixing coefficients $(\kappa_0(t), \kappa_{\mathrm{mid}}(t), \kappa_1(t))$ that interpolate between the source, middle, and target distributions. We adopt a factorized scheduler where the coefficients are defined as

$$\kappa_1(t) = 1 - s(t), \tag{25}$$
$$\kappa_{\mathrm{mid}}(t) = r(t)\, s(t), \tag{26}$$
$$\kappa_0(t) = \big(1 - r(t)\big) s(t), \tag{27}$$

with $s : [0,1] \to [0,1]$ strictly decreasing and $r : [0,1] \to [0,1]$ non-decreasing. Note that this construction implies $\frac{d}{dt} \log \kappa_0(t) \leq \frac{d}{dt} \log \kappa_{\mathrm{mid}}(t), \frac{d}{dt} \log \kappa_1(t)$ (Gat et al., 2024).

In our experiments we use the following parametrization,

$$s(t) = 1 - t^p, \qquad r(t) = t^q, \tag{28}$$

with $p = 2$ and $q = 2/3$. Here $s(t)$ controls the overall decay from source to non-source components, while $r(t)$ redistributes the decaying mass between the middle and the target distributions. This choice yields a unimodal, bell-shaped $\kappa_{\mathrm{mid}}(t)$, peaking at $t^\star = (q/(p+q))^{1/p}$, which for $p = 2$ and $q = 2/3$ gives $t^\star = 0.5$, see Figure 4. Consequently, the middle distribution dominates near the midpoint of the trajectory, aligning with our design goal of exposing the model to acoustically plausible intermediate states. At $t = 0$ and $t = 1$, the path reduces to pure source and pure target distributions, respectively.

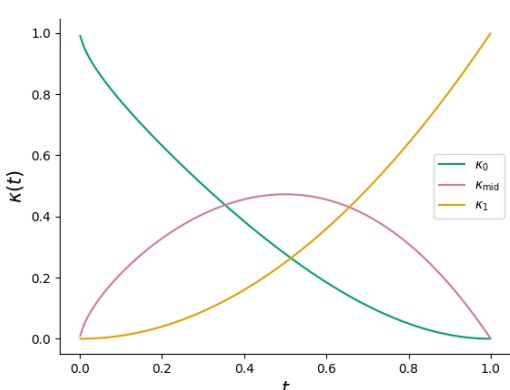

Figure 4: Tri-mixture sampling scheduler.

## C.2 TRAINING AND OPTIMIZATION DETAILS

**Architecture.** Both Drax and Drax-flash uses a Whisper (`large-v3`) encoder ($\sim$ 630M parameters) as the audio encoder. We kept the encoder frozen during training. The decoders uses the DiT (Peebles & Xie, 2023) architecture. The Drax decoder is composed of 16 decoder blocks with 20 attention heads and 1280 hidden dimension with a total of 580M parameters. The Drax-flash decoder contains 4 layers with 20 attention heads and 1280 hidden dimension with a total of 250M parameters. The audio conditioned distribution $p_{\text{mid}}$ contains a single transformer block with cross-attention layer for audio conditioning, together with a projection layer to the vocabulary size, with a total of 28M parameters.

**Optimization.** We train the models using the AdamW optimizer (Loshchilov & Hutter, 2019) with a warmup of 2500 steps and a peak learning rate of $3e - 4$. We use a batch size of 240 and train Drax and Drax-flash for 800K and 250K iterations, respectively.

**Training.** During training we sample $t$ from a uniform distribution on $[0, 1]$. We randomly drop the audio conditioning with probability of 0.1. We adapt the same tokenizer as in Radford et al. (2023), and follow Radford et al. (2023) to prepend the special tokens `<|startoftranscript|><|lang|><|transcribe|><|notimestamps|>`. We allow for random replacement of the language token according to the path $p_t$ with probability 0.2. Furthermore, with probability $p_{\text{prompt}}$, we sample a text prefix from the input utterance which remains unchanged for all $t$. This allows for using Drax for speculative decoding with AR models.

**Datasets.** We train the models using a mix of public datasets covering 8 languages, namely, English, German, Spanish, French, Portuguese, Italian, Chinese, and Japanese, with a total of $\sim$ 15K hours. Specifically, we consider LibriSpeech (LS) Panayotov et al. (2015) (English read audiobooks), Multilingual LibriSpeech (MLS) Pratap et al. (2020) (multilingual read audiobooks), AMI Carletta et al. (2005) (far-field meeting speech), Earnings-22 Rio et al. (2022) (financial earnings calls), VoxPopuli Wang et al. (2021) (multilingual parliamentary speeches), Tedlium Rousseau et al. (2012) (TED talks, prepared speech), CommonVoice-13 Ardila et al. (2019) (crowdsourced read speech), Reazon Fujimoto (2016) (Japanese read speech), and AISHELL Bu et al. (2017) (Mandarin read speech).

**Runtime dataset.** We report RTFx for Drax and compared baselines to measure their runtime efficiency. We note that runtime measurements in ASR models heavily depend on the input audio duration and output sequence length. Therefore, we curate the dataset from LibriSpeech-clean as follows: first, we bin all utterances by duration, ranging from 0 to 30 seconds in 5-second increments; then, we uniformly draw an equal number of samples from each bin. We report summary statistics of ground-truth sequence length and audio duration for the curated dataset in Table 4.

|  | Mean | Std | Min | Max |
|---|---|---|---|---|
| Duration (Sec.) | 12.05 | 6.85 | 1.29 | 28.58 |
| Sequence length (# tokens) | 64.47 | 36.90 | 2.00 | 155.00 |

Table 4: Summary statistics for the curated runtime dataset.

## C.3 SAMPLING

At inference, we use a simple linear scheduler $\kappa_0(t) = 1 - t$ and $\kappa_1(t) = 1 - \kappa_0(t)$. We select $t$ uniformly over $[0, 1]$, i.e. for $K$ NFE set the step size $h = 1/K$, and $t = 1/K, 2/K, ..., K/K$. We sample using the efficient algorithm in Shaul et al. (2024). When evaluation Drax with a single sample we set $\tau = 0.01$, and for generating multiple candidates we use $\tau = 0.1$. We generate sample with a fixed sequence length of $L = 144$. We cache the audio projection and per-block cross-attention K/V tensors so these are computed once per utterance and reused across all generation steps.

## C.4 RUNTIME RESULTS

Runtime-related metrics like RTF and its inverse RTFx were measured on a single L40s GPU. For fair comparison, all methods were evaluated with a batch size of 1, with full-precision and without any compilation.

# D ADDITIONAL RESULTS

## D.1 ACCURACY-EFFICIENCY TRADE-OFF

In Section 5.2, we showed that a key advantage of Drax is the ability to tune the accuracy–runtime trade-off by adjusting the number of NFE and the candidate ensemble size. Here, we expand the evaluation to provide finer control for practitioners: we run a grid search of NFE $\in \{4, 8, 16\}$ and ensemble size $\in \{1, 8, 16\}$. Results are reported in Table 5.

| NFE/Ens. Size | No Scoring | | MBR | | Whisper Score | | Oracle | |
|---|---|---|---|---|---|---|---|---|
|  | WER | RTFx | WER | RTFx | WER | RTFx | WER | RTFx |
| 4/1 | 9.12 | 134.70 | – | – | – | – | – | – |
| 4/8 | – | 64.26 | 7.67 | 62.68 | 7.11 | 44.80 | 6.44 | – |
| 4/16 | – | 38.38 | 7.49 | 36.29 | 6.84 | 28.15 | 6.01 | – |
| 8/1 | 8.61 | 65.59 | – | – | – | – | – | – |
| 8/8 | – | 34.96 | 7.50 | 34.49 | 6.87 | 28.28 | 6.15 | – |
| 8/16 | – | 21.54 | 7.35 | 20.86 | 6.59 | 17.89 | 5.74 | – |
| 16/1 | 8.41 | 32.23 | – | – | – | – | – | – |
| 16/8 | – | 17.87 | 7.41 | 17.75 | 6.73 | 15.95 | 6.03 | – |
| 16/16 | – | 10.98 | 7.28 | 10.80 | 6.49 | 9.94 | 5.64 | – |

Table 5: Effect of number of function evaluations (NFE) and ensemble size on WER and runtime (RTFx) under different scoring methods on the MLS dataset. Candidate generation uses $\tau = 0.1$, and $\tau = 0.01$ when sampling a single transcript.

## D.2 DRAX-FLASH RESULTS

We evaluate Drax-flash using the EN benchmark and the MLS dataset. The results are provided in Tables 6 and 7. In addition, in Figure 5 we visualize the RTFx of Drax-flash, Whisper, and Qwen2-Audio as a function of the transcription sequence length. Drax-flash provides significant improvemnets in runtime.

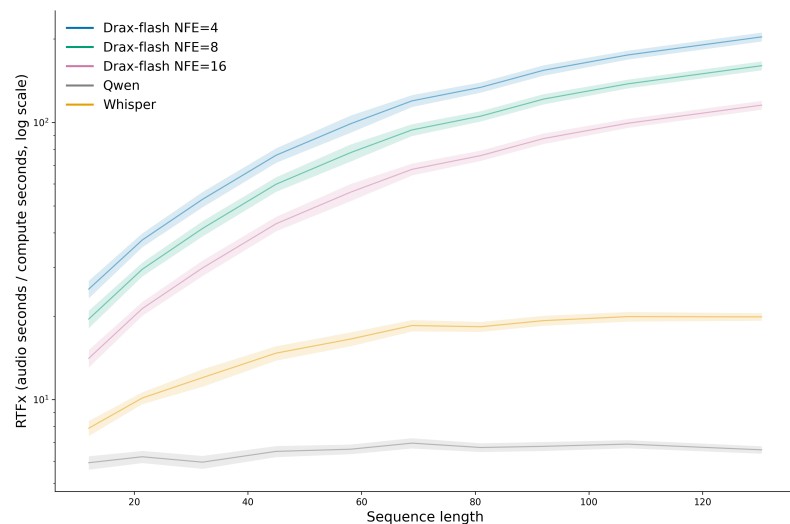

Figure 5: Runtime comparison for Drax-flash.

| | LS Clean | LS Other | AMI | Earnings 22 | VoxPopuli | Tedlium | Average | Params (B) | RTFx↑ |
|---|---|---|---|---|---|---|---|---|---|
| | | | | WER↓ | | | | | |
| Drax-flash | 4.91 | 8.92 | 45.78 | 22.96 | 13.85 | 8.53 | 17.49 | 0.8 | 84.94 |

Table 6: English benchmark, Drax-flash.

### D.3 EXTENDED ENGLISH BENCHMARK RESULTS

We provide additional Drax results for the English benchmark, under different NFE, candidate set size and scoring method setups. The results reported in Table 8 show Drax provide significant control over the accuracy-efficiency trade-off, allowing a user to select its optimal operation point.

### D.4 SAMPLING TEMPERATURE

We also evaluated Drax under different sampling temperatures on the Multilingual LibriSpeech (MLS) benchmark. As expected, increasing temperature leads to more diverse generations but also higher error rates. Lower temperatures (e.g., 0.01-0.1) yield the best WER across languages, while higher values such as 1.0 noticeably degrade performance. Table 9 reports total WERs for each language at different temperatures.

### D.5 EFFECT OF NFE

The number of NFE is a key hyperparameter for NAR based generative models. Here we show that the Drax model achieves high quality sampling with as few as 4-16 sampling steps. Figure 7 show the per-language WER for the MLS and VoxPupoli dataset as a function of NFE. The plot shows that Drax WER drops quickly, with relevantly small improvement for NFE $\geq 4$.

### D.6 SAMPLING WITH AND WITHOUT $p_{\mathrm{MID}}$

As discussed in the Section 3, the intermediate distribution $p_{\mathrm{mid}}$ is introduced as an auxiliary component during training in order to better align the training and inference occupancies. Generally, we do not use $p_{\mathrm{mid}}$ during inference. To verify this design choice, we compare generation with and without $p_{\mathrm{mid}}$ at sampling time (see Appendix B). The results, reported in Figure 8, show that including $p_{\mathrm{mid}}$ during generation consistently hurts performance across datasets. This supports our design choice to treat $p_{\mathrm{mid}}$ as a training-only component.

| | MLS | | | | |
|---|---|---|---|---|---|
| | DE | ES | FR | IT | PT |
| Drax-flash | 13.77 | 10.15 | 14.61 | 20.90 | 20.99 |

Table 7: Drax-flash WER results for the Multilingual LibriSpeech dataset.

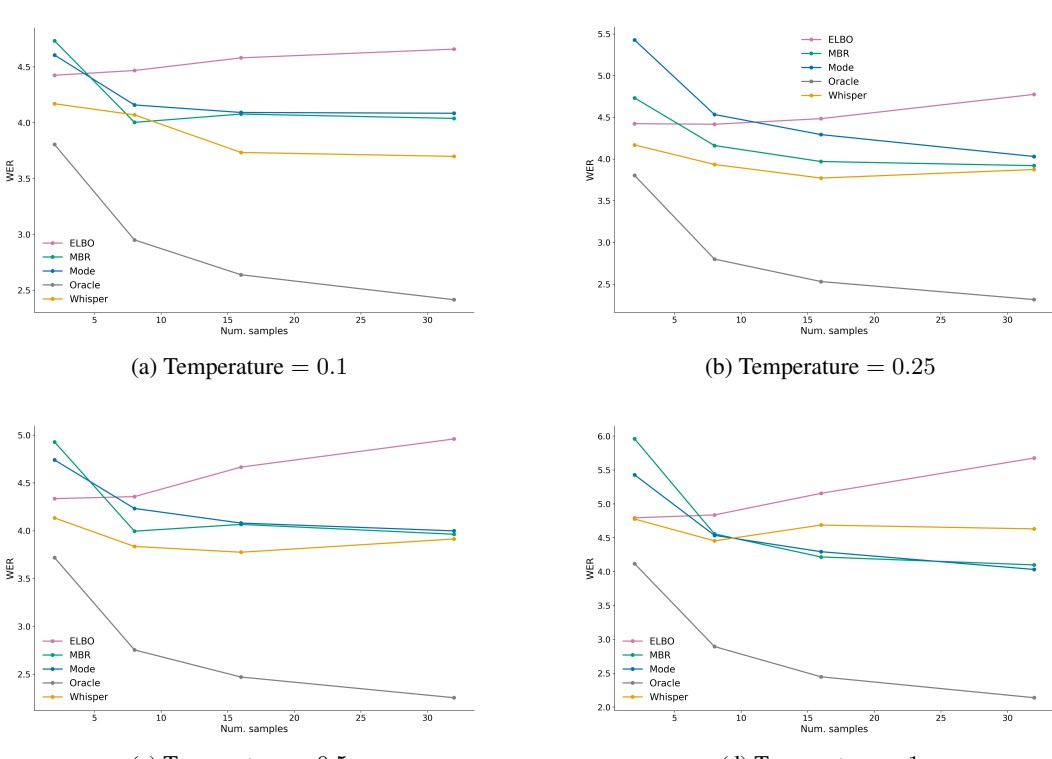

(a) Temperature = 0.1

(b) Temperature = 0.25

(c) Temperature = 0.5

(d) Temperature = 1

Figure 6: Ensemble prediction (candidate scoring) for Tedlium.

### D.7 SAMPLES FROM AUDIO DISTRIBUTION

Table 12 shows an example of samples from the learned middle distribution $p_{\mathrm{mid}}$. We can see that some samples are acoustically plausible but imperfect.

### D.8 GENERATION PATH

To illustrate the decoding dynamics, Table 11 traces Drax's stepwise refinement on a single example. Each row corresponds to a generation step and each column to a token position; cells show the token committed at that step, while "_" denotes positions that remain unchanged from the initial noisy state. The table shows how tokens progressively stabilize across steps, first in shorter blocks and then as longer spans, until the full sequence is recovered.

### D.9 ERROR-TYPE BREAKDOWN

We further report a breakdown of insertion (Ins), deletion (Del), and substitution (Sub) errors for Drax and Whisper (Table 10). Overall, the distribution of error types is quite different between the two models: for Whisper, the largest contribution to WER typically comes from deletions, except on LS Other, where substitutions dominate, whereas for Drax the largest component is usually substitutions. This pattern is consistent across the evaluated datasets and suggests that Drax tends to emit more conservative segmentations, fewer dropped words but more lexical mismatches, while Whisper is more prone to omitting words entirely.

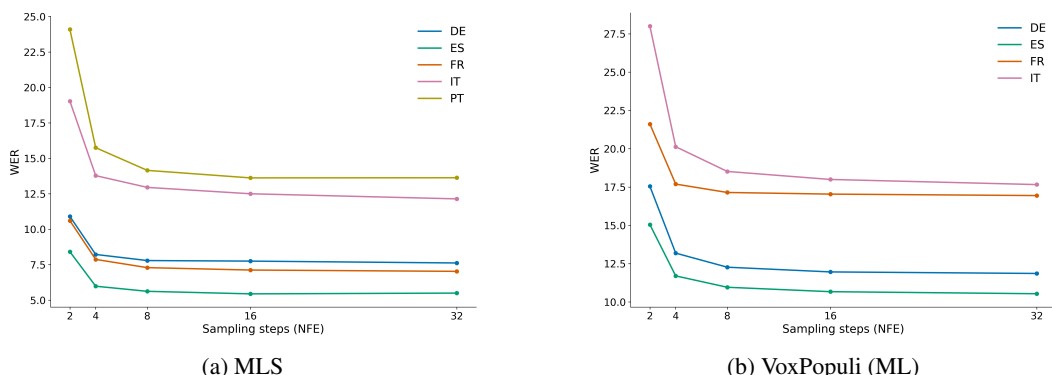

(a) MLS          (b) VoxPopuli (ML)

Figure 7: Effect of sampling steps (NFE).

| | LS Clean | LS Other | AMI | Earnings 22 | VoxPopuli | Tedlium | Average | | |
|---|---|---|---|---|---|---|---|---|---|
| | | | | WER↓ | | | | Params (B) | RTFx↑ |
| Drax (4/1) | 2.9 | 6.3 | 14.3 | 16.3 | 9.0 | 5.7 | 9.1 | 1.2 | 134.7 |
| Drax (8/1) | 2.7 | 5.8 | 13.8 | 15.3 | 8.6 | 5.0 | 8.5 | 1.2 | 65.5 |
| Drax (16/1) | 2.6 | 5.7 | 13.9 | 15.2 | 8.6 | 4.8 | 8.4 | 1.2 | 32.2 |
| Drax (MBR, 8/16) | 2.6 | 5.3 | 13.6 | 14.6 | 8.0 | 4.1 | 8.0 | 1.2 | 20.8 |
| Drax (Whisper, 8/16) | 2.2 | 4.7 | 12.7 | 13.7 | 7.4 | 3.7 | 7.4 | 2.1 | 17.8 |
| Drax (MBR, 16/16) | 2.5 | 5.2 | 13.6 | 14.6 | 7.9 | 4.1 | 7.9 | 1.2 | 10.8 |
| Drax (Whisper, 16/16) | 2.1 | 4.7 | 12.6 | 13.7 | 7.5 | 3.7 | 7.3 | 2.1 | 9.9 |

Table 8: Extended English results for Drax using datasets from the Hugging Face Open ASR benchmark (Srivastav et al., 2023).

### D.10 ROBUSTNESS TO NOISE

Here, assess the noise robustness of our model in comparison with Whisper. Following the original Whisper setup (Radford et al., 2023), we measure WER on LibriSpeech test-clean under white noise and pub noise from the Audio Degradation Toolbox (Mauch et al., 2013), across SNR levels from 40 dB to −10 dB. The results, shown in Figure 9, indicate that our model performs similarly to Whisper at SNRs between 40 and 10, while Whisper demonstrates better robustness at lower SNRs. This behavior is expected, as Whisper was trained on substantially more data (5M hours for Whisper vs. 15K hours for Drax), with greater noise level diversity and in-the-wild audio.

## E LLM USAGE

We used large language models to improve the readability of the manuscript, including grammar and clarity. All research ideas, experiments, and analyses, were conducted and developed by the authors.

## F ETHICS STATEMENT

This work focuses on developing more efficient training and decoding methods for automatic speech recognition. Our models are trained on publicly available multilingual datasets, and we do not foresee immediate ethical risks beyond those already present in ASR systems, such as potential misuse for surveillance or transcription of sensitive conversations. We encourage responsible use of this technology and adherence to relevant privacy and data protection guidelines.

| Temp | DE | ES | FR | IT | PT |
|------|------|------|------|-------|-------|
| 0.01 | 7.75 | 5.44 | 7.12 | 12.50 | 13.62 |
| 0.1 | 7.64 | 5.46 | 7.20 | 12.42 | 13.83 |
| 0.25 | 7.73 | 5.49 | 7.20 | 12.53 | 13.80 |
| 0.5 | 7.93 | 5.61 | 7.39 | 12.80 | 14.32 |
| 1.0 | 8.77 | 6.41 | 8.18 | 14.14 | 16.26 |

Table 9: Total WER of Drax on MLS at different sampling temperatures. Lower values yield better recognition accuracy.

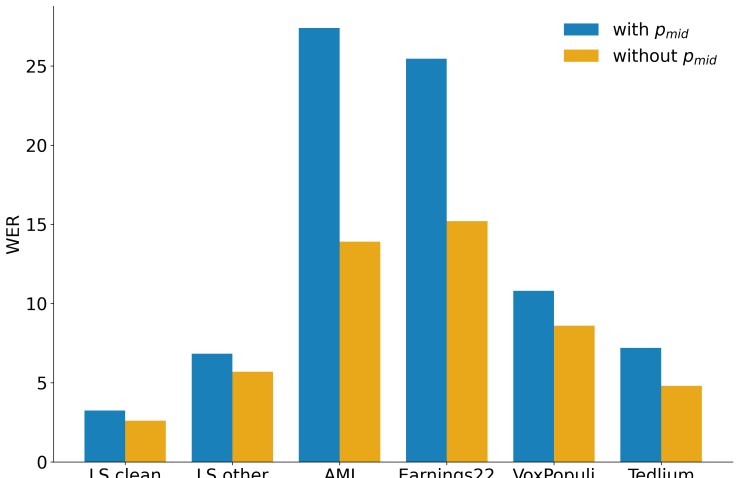

Figure 8: Effect of using $p_{\mathrm{mid}}$ during sampling. Results show that including $p_{\mathrm{mid}}$ at inference degrades accuracy, confirming our choice to use it only during training.

## G    REPRODUCIBILITY STATEMENT

To support reproducibility, we will release pretrained model checkpoints along with the evaluation scripts used in our experiments. All datasets used are publicly available, and we provide full details of hyperparameters, training schedules, and evaluation protocols in the appendix.

|  | LS Other | | | Tedlium | | | Earnings22 | | |
|---|---|---|---|---|---|---|---|---|---|
|  | Ins | Del | Sub | Ins | Del | Sub | Ins | Del | Sub |
| Drax (16/1) | 0.88 | 0.61 | 4.35 | 0.96 | 1.54 | 2.32 | 3.31 | 5.20 | 6.53 |
| Whisper | 0.66 | 0.46 | 2.85 | 0.83 | 2.02 | 1.10 | 2.47 | 4.79 | 3.98 |

Table 10: Breakdown of insertion (Ins), deletion (Del), and substitution (Sub) errors for Drax and Whisper across datasets.

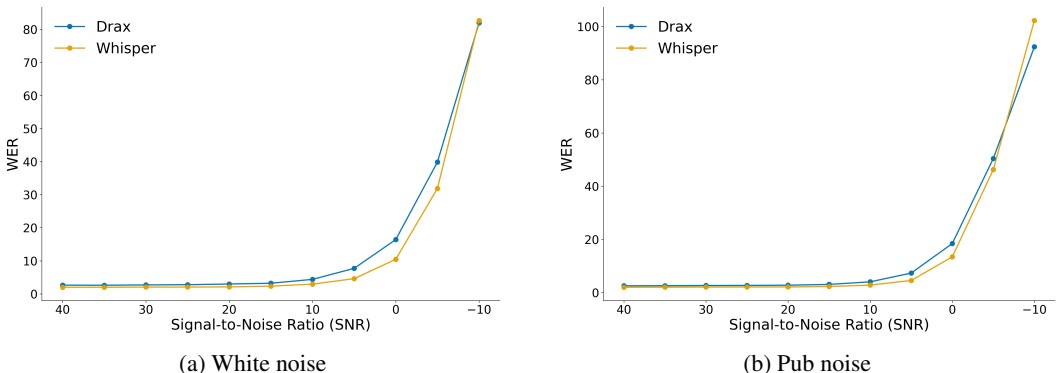

(a) White noise                                  (b) Pub noise

Figure 9: Robustness to noise.

| Step | It | became | the | band's | most | successful | single | worldwide. |
|---|---|---|---|---|---|---|---|---|
| 1 | _ | _ | _ | _ | most | _ | _ | _ |
| 2 | _ | _ | _ | _ | most | _ | _ | _ |
| 3 | _ | _ | _ | _ | most | _ | _ | _ |
| 4 | _ | _ | _ | _ | most | _ | _ | _ |
| 5 | _ | _ | _ | _ | most | _ | _ | worldwide- |
| 6 | _ | _ | _ | _ | most | _ | _ | worldwide- |
| 7 | _ | _ | _ | _ | most | successful | single | worldwide- |
| 8 | _ | _ | _ | _ | most | successful | single | worldwide- |
| 9 | _ | _ | _ | _ | most | successful | single | worldwide- |
| 10 | _ | _ | _ | _ | most | successful | single | worldwide- |
| 11 | It | _ | _ | _ | most | successful | single | worldwide- |
| 12 | It | became | _ | band | most | successful | single | worldwide- |
| 13 | It | became | _ | band's | most | successful | single | worldwide- |
| 14 | It | became | the | band's | most | successful | single | worldwide- |
| 15 | It | became | the | band's | most | successful | single | worldwide. |
| 16 | It | became | the | band's | most | successful | single | worldwide. |

Table 11: Example generation path over 16 steps. Each column corresponds to a token position. "_" marks tokens that remain unchanged from the initial random state at that step.

| Language | Ground Truth | Samples |
|---|---|---|
| EN | And one of the most important | And one the most important |
|  |  | And the one most important |
|  |  | And one of the most important |
| ES | Esto ha pasado y debe pararse de una forma tajante. | esto ha pasado pasado y fair debe per unaarse una una forma |
|  |  | esto ha ha pasadoado y debearsearse uno forma unaante un. |
|  |  | esto ha ha pasado poradoves debe unaarse una formaaj de dos. |
| FR | l'énergie solaire en Europe | 11 lireaireireé europ. |
|  |  | l' sol'èreurlement europe. |
|  |  | l'est sol s.lèreient europe. |

Table 12: Samples from the learned audio distribution $p_{\text{mid}}$.

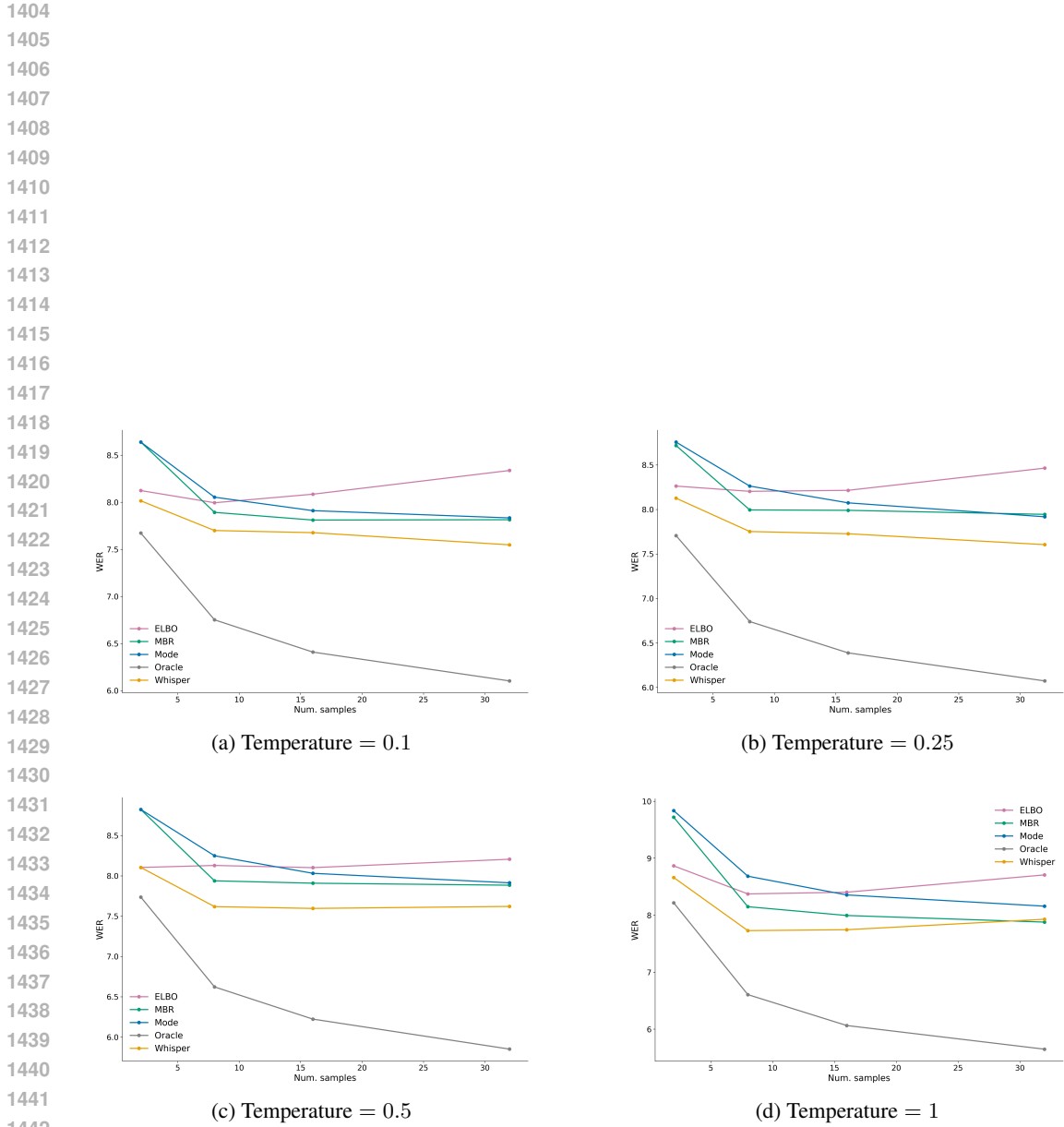

Figure 10: Ensemble prediction (candidate scoring) for VoxPopuli (EN).

