# OpenReview forum: "Drax: Speech Recognition with Discrete Flow Matching"
_ICLR.cc/2026/Conference — Submitted to ICLR 2026_

### Official Review · Reviewer_pZNh · 2025-10-22

**Soundness:** 2
**Presentation:** 3
**Contribution:** 2
**Rating:** 4
**Confidence:** 3

**Summary:**

The paper proposes DRAX, an ASR system trained with Discrete Flow Matching (DFM). It introduces a tri-mixture training path that adds an audio-conditioned intermediate distribution between noise and the target transcript, aiming to better guide the flow during training while keeping non-autoregressive, parallel decoding. Advantages include low latency, a configurable accuracy–latency trade-off via the number of flow steps, support for ensemble/candidate sampling. The authors present results on standard benchmarks and motivate the approach with an occupancy-based generalization bound.

**Strengths:**

The paper introduces a training only tri mixture path with an audio conditioned middle distribution, which (as far as I can tell) has not been used in DFM based ASR or in other DFM applications. The choice is theoretically motivated and appears extendable beyond ASR to other DFM style discrete token tasks.

Non autoregressive, fully parallel decoding with a tunable  number of NFE and the candidate ensemble size  provides a clear accuracy versus latency trade off. However, this advantage stems from the general DFM/NAR framework, not from this paper’s specific path design.

Because the middle distribution is used only during training, the inference pipeline is unchanged with no additional test-time cost. Training overhead is modest, and performance remains comparable over a broad temperature range (Table 8), underlining ease of use in real-world applications

**Weaknesses:**

While the theory is nice, in practice there is no demonstrated link between the theory and the chosen middle path. The theory motivates reducing a training and inference occupancy gap, but there is limited tangible evidence (analytic or empirical) that this specific audio conditioned middle reduces that gap; at best, the paper offers intuition.

I also feel there is a lack of ablations to support the central claim. Given the weak link between the design choice and the theory, the paper should lean on stronger empirics. In particular, there is no version of the main models trained without the middle distribution; only a small, single dataset ablation appears at the end. A extra row in the main tables (1 and/or 2) for the two mixture (no middle) reference would materially strengthen the claim. In particular, since the rationale and modeling presented here seem applicable beyond ASR, a more complete set of experiments (at least within ASR),  including the no middle baseline across multiple datasets, should be the minimum to substantiate the contribution.

Unclear attribution and uneven comparisons within the DFM family. DRAX leverages ensemble prediction, while competing DFM baselines do not appear to be evaluated under the same ensemble regime (correct me if i'm wrong). As a result, improvements may be due to ensembling prediction rather than the proposed method. A comparison with identical numbers of flow steps and fixed candidate counts is needed to fairly attribute gains to the middle path, which again could be achieved by simply running the model without the middle path for the main experiments of the paper.

**Questions:**

Did you try weighting the middle path term (for example, total loss = CDFM loss plus lambda times the middle path loss (equation 9))? Any insights on the impact of such parameter on the results ?

I would suggest to increase figure font sizes; there is enough space to do it and several labels are hard to read.

---

> ### Author Response · Authors · 2025-11-20
> **Response to Reviewer pZNh**
>
> Thank you for the careful reading, the constructive feedback, and the helpful suggestions. We respond to each comment in detail below. We also uploaded a revised version of the paper, with modifications marked in red.
>
> **(Q) Strengthen empirical support for theory and design choice:**
> Thank you. Following the reviewers' comments, we added results for a Drax variant trained with no middle distribution (i.e., with a standard mixture path). This model has the same number of parameters/architecture as the large Drax variant. The results presented here (and in Table 1 of the revised paper), show that adding the middle distribution significantly improves the performance, further strengthening the empirical evidence of our theoretical analysis and motivation:
> | Method                | LS Clean | LS Other | AMI  | Earnings22 | VoxPopuli | Tedlium | Average |
> |-----------------------|----------|----------|------|------------|-----------|---------|---------|
> | DFM                   | 3.4      | 7.0      | 30.8 | 17.6       | 10.5      | 6.9     | 12.7    |
> | Drax                  | 2.6      | 5.7      | 13.9 | 15.2       | 8.6       | 4.8     | 8.4     |
>
> **(Q) Uneven comparisons within the diffusion/DFM family:**
> We respectfully disagree: First, we provide results for Drax with a single candidate (no ensemble), which significantly outperforms all other diffusion/flow-based approaches (see Drax row in Table 1). Second, some of the diffusion-based baselines leverage ensemble prediction (Whisfusion uses 15 candidates, which is comparable with our 16 candidates), or a more complex sampling procedure with multiple NFE (TransFusion, FDDM), for example, TransFusion uses NFE=100, while we use 8 or 16 NFEs.
> Furthermore, we believe the extended results for DFM with the two-mixture path (previous question) further strengthen the empirical evidence of our approach.
>
> **(Q) Weighting the loss term of the middle distribution:**
> We did not experiment with various values for the weighting term. Importantly, the $L_{mid}$ term solely affects the update of the auxiliary network (parameterizes $p_{mid}$) and that of the main flow model. We believe modifying the path design and/or training objective is a viable direction for future improvements, and we leave it for future work.
>
> **(Q) Increase font size in Figures:**
> Thank you. We increased the font size in the revised version of our paper.

---

> > ### Comment · Reviewer_pZNh · 2025-11-24
> > **Reply to authors**
> >
> > Thank you for your reply.
> >
> > I appreciate the effort made by the authors to run the experiment (named DFM) without the middle path.
> >
> > Regarding the comparison with DRAX and other diffusion models, the original paper shows that DRAX without an ensemble does outperform the diffusion models. However, these diffusion models have fewer parameters, and the comparison is limited to LibriSpeech clean/other. In this context, the introduction of the DFM experiment (which I assume was also run without ensemble inference) helps support the claim and shows that the middle-path strategy in DRAX is responsible for the improvement, rather than (only) the ensemble prediction.
> >
> > Although I remain concerned about the weak connection between the design choices and the theory, this point could be discussed in a future paper. Overall, the improvements lead me to increase my score.
> >
> > Best,

---

> > > ### Author Response · Authors · 2025-11-25
> > > **Reply to Reviewer pZNh**
> > >
> > > Thank you for taking the time to reconsider our submission and for indicating your intention to raise the score. We don’t yet see the updated score on our end, so we wanted to kindly check whether this might be due to a technical issue.

---

### Official Review · Reviewer_QEcG · 2025-10-24

**Soundness:** 3
**Presentation:** 2
**Contribution:** 3
**Rating:** 6
**Confidence:** 4

**Summary:**

The paper introduces a discrete Flow Match approach for non-autoregressive (NAR) speech recognition, utilizing a tri-mixture probability path with an audio-conditioned middle distribution.
Experimental results across eight language benchmarks demonstrate competitive accuracy compared to various baselines, offering an effective balance between accuracy and efficiency.

**Strengths:**

The successful integration of discrete Flow Match into NAR Automatic Speech Recognition yields good performance in terms of both accuracy and computational efficiency.

Introduce probability path design within DFW, supported by comprehensive experimental studies and theoretical analysis.

**Weaknesses:**

The approach has not yet been validated on larger scale tasks, nor thoroughly tested for robustness in noisy environments.

There is still insufficient analysis of how to optimize the probability path in ASR, including selecting improved intermediate distributions that can better approximate actual inference dynamics.

**Questions:**

As the Flow Match model can condition on either continuous audio latents or discrete audio tokens, a comparative study examining accuracy, generalisation, and training efficiency would be valuable.

It is advisable to include an analysis of the three error types (Insertion/Deletion/Substitution).

Detailed proofreading is recommended—for example, correcting minor errors such as the duplicate reference observed on line 189.

---

> ### Author Response · Authors · 2025-11-20
> **Response to Reviewer QEcG**
>
> We thank the reviewer for the helpful suggestions, constructive feedback, and the positive assessment. We respond to each comment in detail below. We also uploaded a revised version of the paper, with modifications marked in red.
>
> **(Q) insufficient analysis of how to optimize the probability path:**
> Improving and exploring the training/inference path of flow-matching models is a major design choice and an active area of research [1,2,3]. In our work, we theoretically demonstrate how a more careful training path design can enhance generalization, and empirically show that our path design improves generalization performance.
> We emphasize that analyzing this design choice is non-trivial due to the coupling between training dynamics and inference path.
> Following this comment, we have extended our analysis and included results for a DFM model (with the same number of parameters/architecture as the large Drax variant) trained with the standard 2-way mixture in the revised version. The results show that Drax (trained with our audio-conditioned path design) significantly outperforms the model trained with the standard mixture path. We provide the results here:
>
> | Method                | LS Clean | LS Other | AMI  | Earnings22 | VoxPopuli | Tedlium | Average |
> |-----------------------|----------|----------|------|------------|-----------|---------|---------|
> | DFM                   | 3.4      | 7.0      | 30.8 | 17.6       | 10.5      | 6.9     | 12.7    |
> | Drax                  | 2.6      | 5.7      | 13.9 | 15.2       | 8.6       | 4.8     | 8.4     |
>
>
> **(Q) Robustness to noise analysis:**
> Thank you. Following your comment, we added an analysis for Drax’s robustness to noise in Appendix D.10, following the common procedure from prior works [4]. Our model performs similarly to Whisper at SNRs between 40 and  10, while Whisper demonstrates better robustness at lower SNRs. This behavior is expected, as Whisper was trained on substantially more data (5M hours for Whisper vs. 15K hours for Drax), with greater noise level diversity and in-the-wild audio.
>
> *WER by SNR for White Noise*:
>
> | Model   | 40  | 35  | 30  | 25  | 20  | 15  | 10  | 5   | 0    | -5   | -10   |
> |---------|-----|-----|-----|-----|-----|-----|-----|-----|------|------|-------|
> | Drax    | 2.7 | 2.6 | 2.7 | 2.8 | 3.0 | 3.2 | 4.4 | 7.7 | 16.4 | 39.9 | 82.0  |
> | Whisper | 2.0 | 2.0 | 2.1 | 2.1 | 2.1 | 2.4 | 3.0 | 4.6 | 10.5 | 31.9 | 82.7  |
>
>
> *WER by SNR for Pub Noise:*
>
> | Model   | 40  | 35  | 30  | 25  | 20  | 15  | 10  | 5   | 0    | -5   | -10    |
> |---------|-----|-----|-----|-----|-----|-----|-----|-----|------|------|--------|
> | Drax    | 2.6 | 2.6 | 2.7 | 2.7 | 2.8 | 3.1 | 4.1 | 7.4 | 18.5 | 50.4 | 92.4   |
> | Whisper | 2.0 | 2.1 | 2.1 | 2.1 | 2.2 | 2.3 | 2.9 | 4.6 | 13.5 | 46.3 | 102.3  |
>
>
> **(Q) Conditioning on continuous representation vs. discrete audio tokens:**
> In this work, we opted for using continuous representation, as is the output of the PT encoder. As this is not the focus of our work, we leave further examination of the choice of audio representation, as well as the exact conditioning mechanism, to future works.
>
> **(Q) Add analysis of the three error types (Insertion/Deletion/Substitution):**
> Thank you, we added a breakdown of the WER results in Appendix D.9.
>
> | Model       | LS Other Ins | LS Other Del | LS Other Sub | Tedlium Ins | Tedlium Del | Tedlium Sub | Earnings22 Ins | Earnings22 Del | Earnings22 Sub |
> |-------------|------------------|---------------|---------------|------------------|--------------|--------------|---------------------|-----------------|-----------------|
> | Drax (16/1) | 0.88 | 0.61 | 4.35 | 0.96 | 1.54 | 2.32 | 3.31 | 5.20 | 6.53 |
> | Whisper     | 0.66 | 0.46 | 2.85 | 0.83 | 2.02 | 1.10 | 2.47 | 4.79 | 3.98 |
>
>
> **Reference:**
>
> [1] Flow Matching with General Discrete Paths: A Kinetic‑Optimal Perspective, Shaul et al., ICLR 2025.
>
> [2] Flowing Straighter with Conditional Flow Matching for Accurate Speech Enhancement, Cross et al., 2025.
>
> [3] Elucidating the Design Choice of Probability Paths in Flow Matching for Forecasting, Lim et al., 2024.
>
> [4] Robust Speech Recognition via Large-Scale Weak Supervision, Radford et al., ICML 2023.

---

> > ### Author Response · Authors · 2025-11-27
> > **Follow-up on Discussion and Clarifications**
> >
> > Thank you again for your detailed review and for the helpful feedback. We wanted to kindly follow up to check whether you have any further questions or points you would like us to clarify during the discussion phase. We're happy to provide any additional details that may be useful for your assessment.

---

### Official Review · Reviewer_rrjF · 2025-10-30

**Soundness:** 3
**Presentation:** 3
**Contribution:** 3
**Rating:** 6
**Confidence:** 4

**Summary:**

This paper proposes a novel ASR framework, Drax, based on discrete flow matching. To address the transitions mismatch between training and inference, the audio-conditioned middle distribution is introduced to augment the probability paths. This is motivated by the provided theoretical analysis, which shows the flow matching generalization is controlled by the divergence between training and inference occupancies. Experimental results show that Drax suppasses standard discrete flow matching, and achieves competitive performance with other large scale ASR models, while offering better accuracy-efficiency trade-offs.

**Strengths:**

1) The motivation of this paper is simple and reasonable: standard flow matching only learns transitions from pure noise to target, while ASR inference meets acoustically plausible but imperfect intermediate states, like substitutions, insertions, and deletions. Therefore, the authors improve path designs by augmenting the probability path with the audio-conditioned middle distribution. They also provide theoretical analysis to support that the training-inference path mismatch in flow matching affects generalization ability.
2) The proposed Drax model achieves ASR performance comparable to SOTA large-scale models, while offering improved accuracy-efficiency trade-offs.
3) Good writing, professional academic presentation, and detailed analysis.

**Weaknesses:**

1) As the Drax encoder is a pretrained Whisper (large-v3) model, the authors should also compare to combining a simpler decoder module, such as CTC or AED or Transducer, which is more standard choice in ASR, on the same training data, and see if the proposed method can still get both better accuracy-efficiency trade-offs than these simple methods. I am concerned that, directly comparing to the released Whisper model might not be fair, since it is not trained on the used training data (Appendix C.2). The authors can also finetune the Whisper decoder on these training data for comparison.
2) In Table 1, on Hugging Face Open ASR result, though many popular large-scale models are included, it seems that there are other models achieving better results that Table 1 on https://huggingface.co/spaces/hf-audio/open_asr_leaderboard. It would be better to include a few of them if possible.

**Questions:**

1) In L232-235, "Thus, at test time we are interested in generating directly from the model without relying on the auxiliary component pmid. Concretely, we therefore setαmid ≡ 0 and sample using the same procedure as in the two-way mixture case". I am interested in the results when including the audio-conditioned middle distribution in inference, which should also be reasonable because it align better with the training strategy?
2) In L208-210, "The middle distribution pmid(· | a) is parameterized by an auxiliary network rψ that takes the encoder representation φa as input and outputs per-token categorical distributions." The speech encoder output sequence and target text sequence typically have different lengths. How do the authors handle this? I assume p_mid has the same length of  text sequence.
3) How do the authors determine the target text length in decoding? By EOS tokens or other ways?
4) In Section 5.4, what does "uniform middle" means? If the training paths are only from "audio-conditioned source", is the inference starting from the "audio-conditioned source", or uniform noise source?
5) How about the results if MBR or Whisper-guided scoring are used in standard discrete flow matching model? Would the gain from the audio-conditioned middle distribution disappear?

---

> ### Author Response · Authors · 2025-11-20
> **Response to Reviewer rrjF**
>
> Thank you for the thoughtful, constructive feedback and positive assessment. We respond to each comment in detail below. We also uploaded a revised version of the paper, with modifications marked in red.
>
> **(Q) Comparison with simpler decoder modules (e.g., CTC, AED):**
> We compare Drax to several families of well-established ASR systems, including LLM-based (e.g., Qwen, Voxtral), AED-based (e.g., Whisper, OLMoASR, OWSM), CTC-based (e.g., OWSM-CTC, MMS), and diffusion-based (e.g., TransFusion, Whisfusion).
> In the foundation models landscape, direct comparisons with baselines on a similar setup are generally not possible due to limited access to data (e.g., Whisper-large-v3 was trained with 5M hours of audio), training code, and exact details of the training setup, as well as constraints on available compute.
> Thus, we follow the standard procedure and compare with leading ASR models, similar to prior works.
>
> **(Q) Results when including the audio-conditioned middle distribution in inference:**
> Thank you for this feedback. Following your comment, we provide a comparison for the Drax model w/wo using the audio middle distribution during inference, on several datasets. We include these results in the revised manuscript, in Appendix D.6.
>
> | Model Variant          | LS Clean | LS Other | AMI  | Earnings22 | VoxPopuli | Tedlium |
> |------------------------|----------|----------|------|-------------|-----------|---------|
> | with $p_{mid}$         | 3.2      | 6.8      | 27.4 | 25.4        | 10.8      | 7.2     |
> | w/o $p_{mid}$ (as in paper) | 2.6      | 5.7      | 13.9 | 15.2        | 8.6       | 4.8     |
>
> We show that omitting $p_{mid}$​ during inference leads to better results across datasets. This observation strengthens the design choice in the main paper to operate without the learned middle distribution at inference.
>
> **(Q) Include more baselines in Table 1 (HF leaderboard):**
> The HF leaderboard is continuously updated, with models from both industry and academia. We focused on open models with academic papers/technical reports, and selected state-of-the-art approaches like Qwen2-Audio, Voxtral, and Whisper-large-v3.
> Following your comment, we include an additional baseline, Phi4-multimodal [1], an LLM-based model with ~5B parameters, trained on ~2M hours of proprietary speech-text pairs.
>
> **(Q) Architecture of the auxiliary network for learning $p_{mid}$:**
> The auxiliary network for learning $p_{mid}$ outputs a fixed-sized sequence (similar to the source/target sequences). We use cross-attention for conditioning this sequence on the audio sequence. We have made this clear in the revised manuscript.
>
> **(Q) How to determine the target text length in decoding?**
> Drax outputs a fixed-length sequence of tokens. The output text is padded with EOS tokens.
>
> **(Q) More information on the results in Section 5.4: what is “uniform-middle” and how to sample with an audio-conditioned source?**
> The uniform middle is a tri-mixture path with a uniform middle distribution. Furthermore, when training with an audio-conditioned source distribution, we use this learned distribution to initialize the sequence during sampling.
>
> **(Q) Using candidate scoring with a standard DFM-based model (i.e., no middle distribution):**
> We first note that the results for Whisfusion include their PDD approach, which generates multiple candidates and selects the sequence with the highest score.
> Second, following the reviewer’s comment, we train a DFM model (which is similar to Drax large in terms of architecture/number of parameters) using the standard mixture path (uniform → data). We provide the WER results below. These results show that Drax, trained with the audio-conditioned path, achieves significantly better results on all datasets.
>
>
> | Method                | LS Clean | LS Other | AMI  | Earnings22 | VoxPopuli | Tedlium | Average |
> |-----------------------|----------|----------|------|------------|-----------|---------|---------|
> | DFM                   | 3.4      | 7.0      | 30.8 | 17.6       | 10.5      | 6.9     | 12.7    |
> | DFM (MBR, 8/16)                   | 2.6      | 5.5     |  27.8 | 16.6      | 9.0      | 5.7     | 11.2    |
> | DFM (Whisper, 8/16)              | 2.3     | 5.0      | 20.6 |    15.5  | 8.5     | 5.5     | 9.5   |
> | Drax                  | 2.6      | 5.7      | 13.9 | 15.2       | 8.6       | 4.8     | 8.4     |
> | Drax (MBR, 8/16)      | 2.6      | 5.3      | 13.6 | 14.6       | 8.0       | 4.1     | 8.0     |
> | Drax (Whisper, 8/16)  | 2.2      | 4.7      | 12.7 | 13.7       | 7.4       | 3.7     | 7.4     |
>
>
> **References:**
>
> [1] Phi-4-Mini Technical Report: Compact yet Powerful Multimodal Language Models via Mixture-of-LoRAs, 2025.

---

> > ### Author Response · Authors · 2025-11-27
> > **Follow-up on Discussion and Clarifications**
> >
> > Thank you again for your detailed review and for the helpful feedback. We wanted to kindly follow up to check whether you have any further questions or points you would like us to clarify during the discussion phase. We're happy to provide any additional details that may be useful for your assessment.

---

### Author Response · Authors · 2025-11-20
**General Response**

We thank the reviewers for their time, careful reading, and constructive feedback. We are encouraged that the reviewers found the paper well written and professionally presented (rrjF, pZNh) and appreciated the clarity of motivation and design of the proposed approach (rrjF). Reviewers highlighted the novelty of introducing an audio-conditioned middle distribution within discrete flow matching (rrjF, QEcG, pZNh), noting its simplicity, intuitive grounding, and potential applicability beyond ASR (pZNh). The reviewers acknowledged the strength of the theoretical analysis, particularly the occupancy-based generalization perspective (rrjF, QEcG). They also highlighted the strong empirical results, including competitive ASR accuracy, a favorable accuracy-efficiency trade-off, and the advantages of non-autoregressive, fully parallel decoding (rrjF, QEcG, pZNh). Reviewers also noted the breadth of experiments across eight benchmarks (QEcG) and the detailed analysis in the empirical section (rrjF).

Below, we provide detailed, point-by-point responses to each reviewer’s questions and concerns. We also uploaded a revised version of the paper, with modifications marked in red for ease of reference.

---

### Author Response · Authors · 2025-12-02
**Post-discussion summary for AC**

We thank the AC and reviewers for their careful evaluation and constructive feedback. Our work introduces Drax, a discrete flow–matching multilingual ASR model with an audio-conditioned middle distribution that enables fully parallel, non-autoregressive decoding with a favorable accuracy-efficiency trade-off.

In the revision, we directly addressed the main concerns:

* We added a strong DFM baseline without the middle distribution, matched in architecture and parameters, and showed that our audio-conditioned path consistently improves WER across all datasets.
* We clarified fairness within the diffusion/DFM family by reporting Drax results both with a single candidate and with ensembles, and by adding DFM variants with MBR and Whisper-guided scoring.
* We expanded the empirical analysis with robustness-to-noise experiments, insertion/deletion/substitution breakdowns, additional baselines (e.g., Phi-4-Mini), and clearer architectural details and figures.

We are encouraged that, during the discussion phase, **all three reviewers converged to positive recommendations (scores 6, 6, 6)**, and one reviewer explicitly stated that they were raising their score after considering the new experiments and clarifications. The other reviewers did not further engage in the written discussion, but also did not express any new reservations beyond their already positive assessments. Given this consensus, together with the novelty of audio-conditioned training paths in discrete flow matching, our theoretical contribution, and the strengthened empirical support, we respectfully ask that you weigh the updated assessments and full discussion record when considering Drax for acceptance.

---

### Meta-Review · Area_Chair_weBw · 2026-01-04

**Summary:**

The paper proposes a new model for non-autoregressive ASR based on discrete flow matching. The reviewers were positive on the motivation, and the approach itself, and the ASR quality, which is close to the SOTA when using auto-regressive models.

Main issues pointed out by the reviewers:
1. Fair comparison (training baselines on the same data and architecture), and / or missing SOTA baselines.
2. Missing results in realistic conditions (noise-robustness).
3. Additional analysis on how to optimize probability path during training / inference.
4. Lack of results to support the main claim (of using middle distribution during training to improve inference quality), and the weak connection between the design choice of the middle distribution.

**Reviewer Concerns:**

When comparing with results from the literature, unless the models are trained on the same data or achieves significant gains over SOTA, it’s hard to argue whether the presented method is an improvement. Non-AR based techniques have been proposed in the past, so the novelty here is limited. And given the results are only close (and in a lot of cases, behind) SOTA, and not surpassing it, the paper is really borderline.

Main issues that were not fully addressed, and which, in my opinion, makes this really borderline: The authors did not include results when baseline are trained on similar data, performance in noisy conditions (and other sets) are significantly worse compared to SOTA.

More details below:

1. The authors argue that they have provided baselines from the literature, and that a direct comparison is not possible. But surely, the authors can finetune whisper using the same training data to see if the results are comparable? The authors added a few more results from huggingface, which the reviewer pointed out, and they are significantly better than the Drix.
2. The authors compare with whisper at various SNRs; whisper works much better but the authors argue it’s because of the training data. But this is unsatisfying, since the authors also use a separate training set and compare with results from the literature trained on different sets.
3. The authors argue that this is still an active area, and the focus of the work is show that performance can be improved by better path design during training. They also included results for a more standard DFM model. But in the results, it is surprising to see AMI improve so much (30.8 -> 13.9), when the other models only improved marginally.
4. The reviewer agreed that the DFM results added in rebuttal answers this satisfactorily.

**Reviewer Scores:**

Reviewer rrjF (Zengwei Yao): 6 -> 4/6

Reviewer QEcG (Lei He): 6 -> 4/6

Reviewer pZNh (Yacouba Kaloga): 4 -> 6

---

### Decision · Program_Chairs · 2026-01-26

Reject